# DHODH inhibition enhances the efficacy of immune checkpoint blockade by increasing cancer cell antigen presentation

Nicholas J Mullen[1], Surendra K Shukla[2], Ravi Thakur[2], Sai Sundeep Kollala[1], Dezhen Wang[1], Nina Chaika[1], Juan F Santana[3], William R Miklavcic[1], Drew A LaBreck[2], Jayapal Reddy Mallareddy[1], David H Price[3], Amarnath Natarajan[1], Kamiya Mehla[2], David B Sykes[4,5], Michael A Hollingsworth[1], Pankaj K Singh[1,2,6]*

[1]Eppley Institute for Research in Cancer and Allied Diseases, University of Nebraska Medical Center, Omaha, United States; [2]Department of Oncology Science, University of Oklahoma Health Sciences Center, Oklahoma City, United States; [3]Department of Biochemistry and Molecular Biology, University of Iowa, Iowa City, United States; [4]Center for Regenerative Medicine, Massachusetts General Hospital, Boston, United States; [5]Harvard Stem Cell Institute, Cambridge, United States; [6]OU Health Stephenson Cancer Center, University of Oklahoma Health Sciences Center, Oklahoma City, United States

*For correspondence:
pankaj-singh@ouhsc.edu

**Abstract** Pyrimidine nucleotide biosynthesis is a druggable metabolic dependency of cancer cells, and chemotherapy agents targeting pyrimidine metabolism are the backbone of treatment for many cancers. Dihydroorotate dehydrogenase (DHODH) is an essential enzyme in the de novo pyrimidine biosynthesis pathway that can be targeted by clinically approved inhibitors. However, despite robust preclinical anticancer efficacy, DHODH inhibitors have shown limited single-agent activity in phase 1 and 2 clinical trials. Therefore, novel combination therapy strategies are necessary to realize the potential of these drugs. To search for therapeutic vulnerabilities induced by DHODH inhibition, we examined gene expression changes in cancer cells treated with the potent and selective DHODH inhibitor brequinar (BQ). This revealed that BQ treatment causes upregulation of antigen presentation pathway genes and cell surface MHC class I expression. Mechanistic studies showed that this effect is (1) strictly dependent on pyrimidine nucleotide depletion, (2) independent of canonical antigen presentation pathway transcriptional regulators, and (3) mediated by RNA polymerase II elongation control by positive transcription elongation factor B (P-TEFb). Furthermore, BQ showed impressive single-agent efficacy in the immunocompetent B16F10 melanoma model, and combination treatment with BQ and dual immune checkpoint blockade (anti-CTLA-4 plus anti-PD-1) significantly prolonged mouse survival compared to either therapy alone. Our results have important implications for the clinical development of DHODH inhibitors and provide a rationale for combination therapy with BQ and immune checkpoint blockade.

## eLife assessment

This **important** study reports a novel mechanism linking DHODH inhibition and subsequent pyrimidine nucleotide depletion with upregulation of cell surface MHC I in cancer cells. The in vitro mechanistic data are **compelling**, with rigorous methodology and validation across multiple cell lines. The

authors also provide in vivo evidence for additive effects of DHODH inhibitors and immune check-point blockade. However, the in vivo assessments of the functional relevance of this mechanism remain **incomplete**, requiring additional analyses to fully substantiate the conclusions made.

## Introduction

Deranged cellular metabolism is a universal feature of cancer cells (*Warburg, 1956*; *Hanahan and Weinberg, 2011*). One particularly cancer-essential metabolic aberration is the hyperactive synthesis and utilization of nucleotide triphosphates; this phenotype is a critical driver of cancer cell malignant behaviors, including uncontrolled proliferation, evasion of the host immune response, metastasis to distant organs, and resistance to antineoplastic therapy (*Mullen and Singh, 2023*). The de novo pyrimidine biosynthesis pathway, which generates pyrimidine nucleotides from aspartate and glutamine, is consistently hyperactive in cancer cells and druggable by clinically approved inhibitors (*Wang et al., 2021*). Dihydroorotate dehydrogenase (DHODH) catalyzes the fourth step in this pathway and is essential for de novo pyrimidine synthesis. DHODH inhibitors have shown robust preclinical anti-cancer activity across diverse cancer types (*Shukla et al., 2017*; *Christian et al., 2019*; *Sykes et al., 2016*; *Wang et al., 2019*; *Koundinya et al., 2018*; *Santana-Codina et al., 2018*; *Brown et al., 2017*; *Mathur et al., 2017*; *Li et al., 2019a*; *Bajzikova et al., 2019*) and have recently entered clinical trials for multiple hematological cancers (NCT04609826 and NCT02509052). Although there is a vast literature on DHODH inhibitors dating back to the early 1990s, and despite the 'rediscovery' of DHODH in recent years as a critical cancer cell metabolic dependency, important questions about the cellular response to DHODH inhibition remain unanswered.

While combination chemotherapy is highly effective and potentially curative against certain cancers (e.g., Hodgkin lymphoma, testicular cancer, childhood leukemia, and others), many common malignancies are refractory to chemotherapy (e.g., lung cancer, pancreatic cancer, colorectal cancer, etc.) (*Falzone et al., 2018*). In some chemotherapy-refractory cancers (most prominently melanoma, mismatch repair-deficient colorectal cancer, bladder cancer, and non-small cell lung cancer), immunotherapeutic strategies have demonstrated strong efficacy and led to durable remissions in a subset of patients (*Vaddepally et al., 2020*). The efficacy of immunotherapy agents is dependent on multiple factors, including tumor antigen presentation, limited immune cells in the tumor milieu, and T-cell activation status (*Mundry et al., 2020*; *Waldman et al., 2020*). Adoptive cell therapies and immune checkpoint blockade (ICB) can address the issues of limited immune cell recruitment into tumors and limited T-cell activation, respectively. However, adequate antigen presentation by tumor cells is still required for immunotherapy efficacy, which relies on T-cell-mediated adaptive immunity.

The antigen presentation pathway (APP) mediates the presentation of endogenous peptide antigens to CD8 T-cells via MHC class I (MHC-I). This pathway entails the degradation of cellular proteins into small peptides by the proteasome, the import of these peptides into the endoplasmic reticulum by transporter associated with antigen presentation proteins (*TAP1* and *TAP2*), and the loading of these peptides into the MHC-I complex, which consists of a heavy chain (encoded by *HLA-A*, *HLA-B*, or *HLA-C*) and a light chain (encoded by *B2M*) (*Pishesha et al., 2022*). APP genes are often downregulated in cancer cells, and this impedes the recognition of immunogenic MHC-I restricted cancer cell antigens by infiltrating T-cells (*Cornel et al., 2020*). Antigen presentation and T-cell recognition are crucial for T-cell-mediated killing of cancer cells (*Dhatchinamoorthy et al., 2021*; *Han et al., 2019*; *Zaretsky et al., 2016*), and forced MHC-I expression enhances immunotherapy efficacy in preclinical models (*Yamamoto et al., 2020*; *Goel et al., 2017*; *Kalbasi et al., 2020*; *Gu et al., 2021*). Furthermore, high tumoral expression of MHC-I, MHC-II, and other APP genes correlates with better overall survival in patients with melanoma treated with ICB therapies (*Rodig et al., 2018*; *Liu et al., 2019*; *Grasso et al., 2020*; *Shklovskaya et al., 2020*).

While previous reports have shown that pyrimidine nucleotide depletion triggers the expression of innate immunity-related genes and induces an interferon-like response (*Luthra et al., 2018*; *Lucas-Hourani et al., 2013*; *Sprenger et al., 2021*), the role of pyrimidine starvation in antigen presentation has not been reported. Herein, we report that DHODH inhibition induces the robust upregulation of APP genes and increases tumor cell antigen presentation via MHC-I. We further explored the mechanism and functional consequences of DHODH inhibitor-mediated APP induction in cancer.

## Results

### Brequinar induces upregulation of MHC-I and APP genes

We examined gene expression changes following transient or prolonged DHODH inhibition by culturing human pancreatic ductal adenocarcinoma cell lines S2-013 and CFPAC-1 in the presence of brequinar (BQ) at two different doses for 16 hr and for a 2-week duration (*Figure 1A*). Gene set enrichment analysis (GSEA) using Hallmark and KEGG gene sets from MSigDB (*Liberzon et al., 2011*; *Subramanian et al., 2005*) revealed 17 gene sets that were significantly upregulated (FDR q < 0.25) across both cell lines following 2-week BQ exposure (*Figure 1B*). Twelve of these gene sets (highlighted in purple) are ontologically related to antigen presentation and contain MHC class I, MHC class II, and/or APP genes such as *TAP1* in the leading edge. Certain gene sets, such as allograft rejection (KEGG), graft versus host disease (KEGG), and antigen processing and presentation (KEGG), are composed almost entirely of APP genes (*Figure 1C*). Heatmap analysis showed that APP genes were robustly upregulated in a dose- and duration-dependent manner in CFPAC-1 (*Figure 1D*) and S2-013 (*Figure 1—figure supplement 1A*) cells. The effect size was generally smaller for S2-013 cells, likely because they are resistant to DHODH inhibition due to efficient nucleoside salvage, as we previously reported (*Mullen et al., 2023*). Publicly available RNA-seq data from human A375 melanoma cells treated with the clinically approved DHODH inhibitor teriflunomide (*Tan et al., 2016*) corroborated our findings, as teriflunomide caused a rapid (within 12 hr) and duration-dependent increase in MHC-I/II and APP transcript levels (*Figure 1E*).

We validated these gene expression changes in CFPAC-1 cells by RT-qPCR (*Figure 1—figure supplement 1B*) and then performed RT-qPCR to assess the mRNA levels of genes coding for MHC-I across a panel of human cancer cell lines treated with BQ for 24 hr (*Figure 1F*). This confirmed that MHC-I heavy chain transcripts (*HLA-A*, *HLA-B*, and *HLA-C*) are consistently upregulated in response to BQ across diverse cancer types (*Figure 1F*). To optimize conditions for in vivo studies, we tested the long-term response and observed that 2-week BQ treatment of B16F10 murine melanoma cells also caused dramatic APP gene upregulation (*Figure 1—figure supplement 1C*). Flow cytometry confirmed a marked increase in cell surface MHC-I levels in nonpermeabilized live CFPAC-1 (*Figure 1G*) and B16F10 (*Figure 1H*) cells following a 2-week BQ treatment, confirming that transcriptional upregulation of APP genes results in greater cell surface antigen presentation.

In parallel, we confirmed pyrimidine nucleotide depletion upon treatment with BQ at different doses by performing metabolomics analysis of CFPAC-1 and B16F10 cells following BQ treatment. The results demonstrated a rapid (8 hr treatment) and dose-dependent accumulation of dihydroorotate and N-carbamoyl-aspartate (upstream of DHODH) as well as depletion of pyrimidine nucleotides UTP and CTP (*Figure 1I and J*) and other pyrimidine species (*Figure 1—figure supplement 1D and E*). These results confirm that on-target DHODH inhibition and resultant pyrimidine nucleotide depletion correlates with the transcriptional induction of APP genes and enhanced antigen presentation via MHC-I.

### BQ-mediated APP induction depends on pyrimidine nucleotide depletion

To confirm that BQ- or teriflunomide-mediated APP induction was specifically caused by DHODH inhibition (i.e., on-target effect), we asked whether the effect could be reversed by restoring pyrimidine nucleotides in B16F10 mouse melanoma cells. As we previously observed (*Mullen et al., 2023*), media supplementation with uridine rescued cell viability (*Figure 2A*) and pyrimidine levels (*Figure 2B*) following BQ treatment and partially rescued viability following teriflunomide treatment (*Figure 2—figure supplement 1A*). Uridine supplementation likewise blocked mRNA induction of mouse MHC-I transcripts (*H2-Db*, *H2-Kb*, and *B2m*), as well as *Nlrc5* (a major MHC-I transcriptional coactivator) and *Tap1* (required for peptide import into the ER, a key step in MHC-I antigen presentation) by BQ (*Figure 2C*) or teriflunomide (*Figure 2—figure supplement 1B*), while uridine alone had no effect (*Figure 2—figure supplement 1C*). This same phenotype was observed in HCT116 human colorectal cancer cells (*Figure 2—figure supplement 1D*). Concordantly, cell surface MHC-I upregulation by BQ or teriflunomide (24 hr treatment) was abrogated by uridine supplementation (*Figure 2D*), while uridine alone again had no effect (*Figure 2—figure supplement 1E*). These results demonstrate that DHODH inhibitor-mediated APP induction is caused by pyrimidine nucleotide depletion.

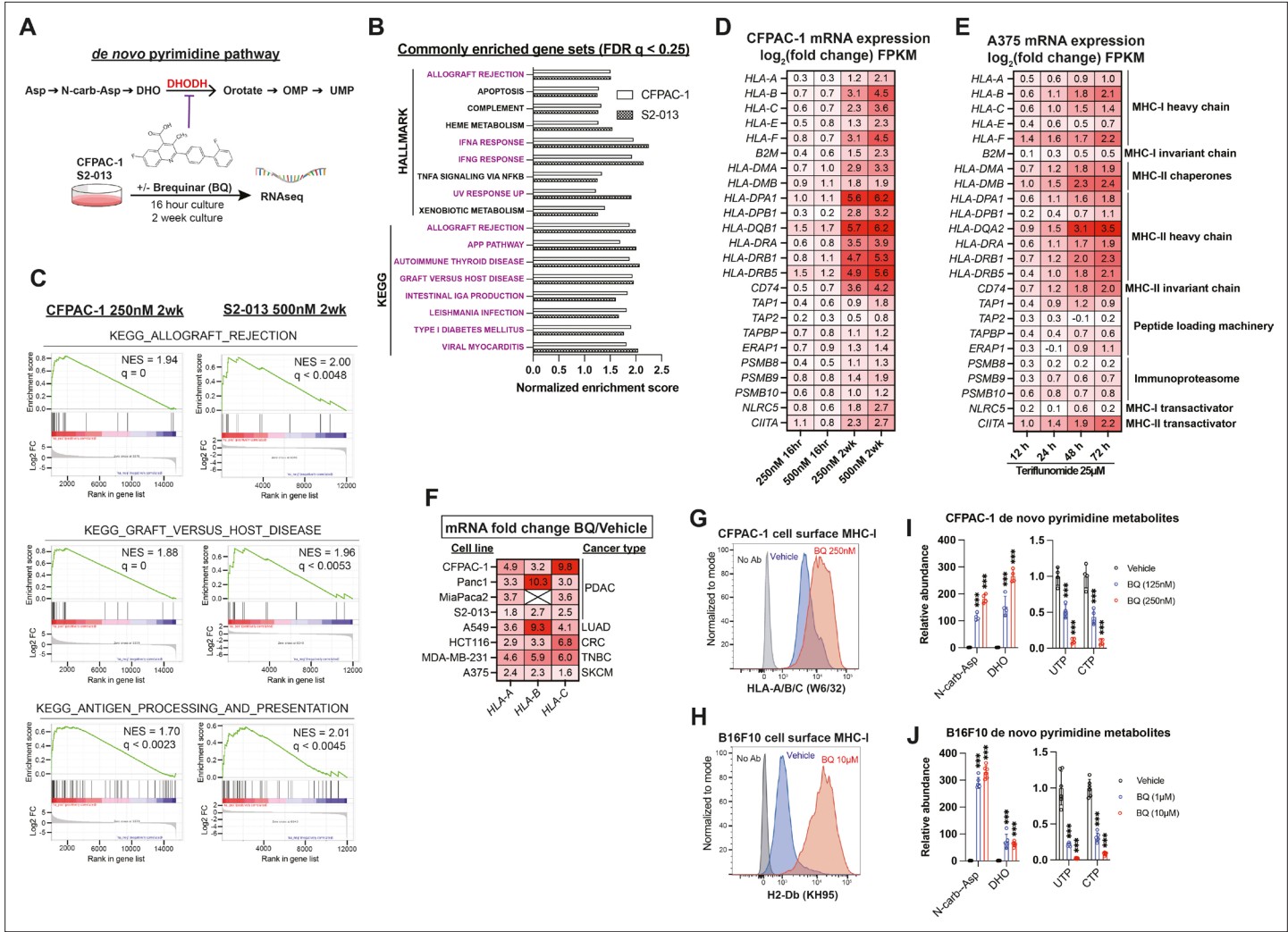

**Figure 1.** Brequinar (BQ) induces mRNA expression of antigen presentation pathway genes and upregulates cell surface MHC-I in diverse cancer cell lines. (**A**) Schematic of RNA sequencing experiment for panels (**B–D**), with de novo pyrimidine pathway shown to highlight the role of DHODH. (**B**) Normalized enrichment scores for gene sets commonly enriched (FDR q < 0.25) in S2-013 and CFPAC-1 cells following 2-week BQ treatment (250 nM for CFPAC-1; 500 nM for S2-013), as assessed by gene set enrichment analysis (GSEA). (**C**) GSEA plots for indicated gene sets following 2-week BQ treatment of CFPAC-1 (left) or S2-013 (right) cells at the indicated doses. (**D**) Heatmap showing log2 fold change mRNA expression measured by RNA sequencing of APP genes in CFPAC-1 cells treated with BQ for indicated dose and duration. (**E**) Heatmap showing log2 fold change mRNA expression measured by RNA sequencing for APP genes in A375 melanoma cells treated with the DHODH inhibitor teriflunomide (25 µM) for indicated durations, data extracted from *Tan et al., 2016*. (**F**) RT-qPCR quantification of *HLA-A*, *HLA-B*, and *HLA-C* mRNA levels in cancer cell lines after 24 hr BQ treatment. Numbers represent fold change relative to vehicle control for each cell line. Data are representative of at least three independent experiments. *HLA-B* was not detectable in MiaPaCa2 cells. (**G, H**) Flow cytometry analysis of cell surface MHC-I in live CFPAC-1 (**G**) or B16F10 (**H**) cells following 10-day treatment with BQ (250 nM for CFPAC-1 and 10 µM for B16F10). (**I, J**) Liquid chromatography-tandem mass spectrometry metabolomics quantification of de novo pyrimidine pathway metabolites in CFPAC-1 (**I**) or B16F10 (**J**) cells following 8 hr BQ treatment at indicated doses. Data represent mean ± SD of four (CFPAC-1) or six (B16F10) biological replicates. ***p<0.001 by one-way ANOVA with Dunnett's multiple-comparison test.

The online version of this article includes the following source data and figure supplement(s) for figure 1:

**Source data 1.** Source data for gene set enrichment analysis, RT-qPCR, and metabolomics experiments shown in *Figure 1* and *Figure 1—figure supplement 1*.

**Figure supplement 1.** Brequinar (BQ) treatment upregulates APP genes and depletes pyrimidine nucleotides.

To further validate this finding, we assessed MHC-I heavy chain mRNA levels in S2-013 cells with DHODH deletion (sgDHODH). We have previously demonstrated that these cells require exogenous uridine for viability and experience profound pyrimidine depletion (>95% depletion of UTP and CTP) after 8 hr incubation in nucleoside-free media (*Mullen et al., 2023*). After growing these

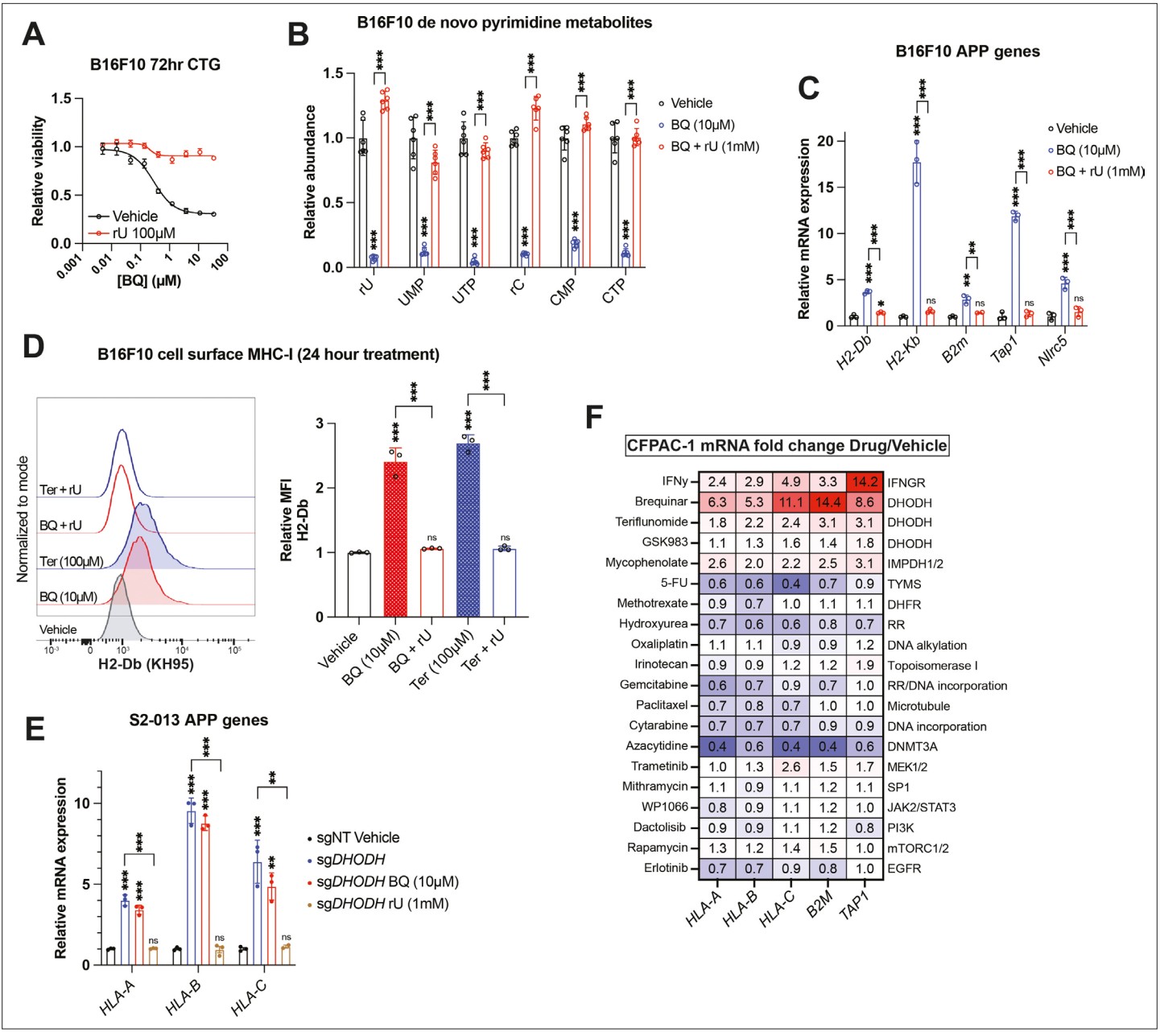

**Figure 2.** Brequinar (BQ)-mediated APP induction requires pyrimidine nucleotide depletion. (**A**) Dose–response cell viability experiment in B16F10 cells treated with BQ ± uridine (100 µM) for 72 hr. Data represent mean ± SEM of three biological replicates. One representative result of three independent experiments is shown. (**B**) Quantification of pyrimidine metabolites following 24 hr treatment of B16F10 cells with vehicle, BQ (10 µM), or BQ + uridine (1 mM). Data represent mean ± SD of six biological replicates. ***p<0.001 by one-way ANOVA with Tukey's multiple-comparison test. (**C**) RT-qPCR quantification of mRNA levels for indicated APP genes in B16F10 cells following 24 hr treatment with BQ (10 µM) ± uridine (1 mM). Data represent mean ± SD of three technical replicates. One representative result of three independent experiments is shown. **p<0.01, ***p<0.001, and 'ns' p>0.05 by one-way ANOVA with Tukey's multiple-comparison test. (**D**) Left: flow cytometry analysis of cell surface MHC-I (H2-Db) on live B16F10 cells following 24 hr treatment with indicated agents (BQ 10 µM, teriflunomide 100 µM, uridine 1 mM). Right: quantification of H2-Db mean fluorescence intensity normalized to vehicle control. Data represent mean ± SD of three independent experiments. ***p<0.001 and 'ns' p>0.05 by one-way ANOVA with Tukey's multiple-comparison test. (**E**) RT-qPCR quantification of mRNA levels for indicated APP genes in S2-013 cells with DHODH knockout (sgDHODH) or non-targeting control vector (sgNT) treated with indicated agents for 72 hr. Data represent mean ± SD of four determinations. One representative result of three independent experiments is shown. **p<0.01, ***p<0.001, and 'ns' p>0.05 by one-way ANOVA with Tukey's multiple-comparison test. (**F**) RT-qPCR quantification of mRNA levels for indicated APP genes in CFPAC-1 cells following 72 hr treatment with indicated agents. Numbers in the heatmap represent mean fold change versus vehicle control with four determinations.

The online version of this article includes the following source data and figure supplement(s) for figure 2:

*Figure 2 continued on next page*

*Figure 2 continued*

**Source data 1.** Source data for cell viability, metabolomics, RT-qPCR, and flow cytometry experiments shown in *Figure 2* and *Figure 2—figure supplement 1*.

**Figure supplement 1.** Brequinar and teriflunomide cause MHC-I upregulation by pyrimidine nucleotide depletion.

cells with supplemented uridine (1 mM), we withdrew exogenous nucleosides by changing to new media containing 10% dialyzed fetal bovine serum (FBS). After 72 hr exposure to nucleoside-free media, sgDHODH cells upregulated *HLA-A*, *HLA-B*, and *HLA-C*, and this was reversed by adding back uridine (*Figure 2E*). Importantly, treatment with BQ did not further increase MHC-I mRNA expression (*Figure 2E*, compare blue and red bars). Together with our other data, these results indicate that BQ-mediated APP induction is an on-target phenomenon with respect to DHODH inhibition.

Since uridine addback rescued BQ- and teriflunomide-mediated loss of viability (*Figure 2A*, S2A), we queried whether BQ-mediated APP induction was caused by pyrimidine depletion per se, or if it was the result of some nonspecific downstream consequence of pyrimidine starvation, such as DNA damage or loss of cell viability. To address this, we screened a panel of genotoxic chemotherapy agents and small molecule inhibitors for their ability to induce APP genes following 72 hr exposure at previously determined cytotoxic doses in CFPAC-1 cells (*Figure 2F*). Besides interferon gamma (a positive control), BQ, teriflunomide, and GSK983 (another DHODH inhibitor), the only agent that induced APP gene transcription in this assay was mycophenolate, a clinically approved inhibitor of the de novo GTP synthesis enzymes inosine monophosphate dehydrogenase 1 and 2 (IMPDH1/2). The effect of mycophenolate on APP gene expression was subsequently validated in B16F10 cells (*Figure 2—figure supplement 1F*), demonstrating that either purine or pyrimidine nucleotide depletion can induce cancer cell APP mRNA expression in vitro.

The other drugs screened included nucleotide synthesis inhibitors (5-fluorouracil, methotrexate, gemcitabine, and hydroxyurea), DNA damage inducers (oxaliplatin, irinotecan, and cytarabine), a microtubule targeting drug (paclitaxel), a DNA methylation inhibitor (azacytidine), and other small molecule inhibitors (*Figure 2F*). While we cannot rule out the possibility that these agents induce APP transcription in other cell lines or under other dose/duration conditions, the inertness of these compounds (with respect to APP gene expression) in our screen suggests that BQ-mediated APP induction in CFPAC-1 cells is not a general phenomenon that occurs downstream of DNA damage or some other response to therapy-induced stress.

## BQ-mediated APP induction does not depend on canonical APP transcriptional regulators

To elucidate the molecular pathway leading to APP induction downstream of pyrimidine depletion, we extended our findings to HEK-293T cells, which also display rapid (within 4 hr) transcriptional induction of MHC-I upon BQ treatment (*Figure 3—figure supplement 1A*). Reasoning that the mechanism of this phenomenon in HEK-293T cells is less likely to involve idiosyncratic genetic aberrations than in cancer cell lines, we chose to conduct our initial mechanistic studies in this system and then extend our findings to cancer cell lines if possible.

We used a candidate-based chemical biology screening approach to ask if drugs targeting suspected pathways might block BQ-mediated APP induction in HEK-293T cells. We first interrogated pathways that are known to control MHC/APP expression, including IFN-JAK-STAT (*Zhou, 2009*), NF-κB (*Gu et al., 2021*; *Dejardin et al., 1998*), and cGAS-STING-TBK1 (*Li et al., 2019b*). Neither ruxolitinib (a JAK1/2 inhibitor with activity against STAT3) nor GSK8612 (a TBK1 inhibitor) (*Thomson et al., 2019*), nor TPCA-1 (an IKK2 inhibitor) (*Podolin et al., 2005*) abrogated BQ-mediated APP induction (*Figure 3A*), despite blocking APP induction downstream of poly(dA:dT) and interferon gamma (*Figure 3—figure supplement 1B*) as expected. This indicates that these canonical regulators of MHC/APP expression are dispensable for APP induction downstream of DHODH inhibition.

Interestingly, the IKK2 inhibitor BMS-345541 (*Burke et al., 2003*) mostly abrogated BQ-mediated APP induction (*Figure 3A*). BMS-345541 effectively blocked BQ- and Ter-mediated APP induction at concentrations of 10 μM and 40 μM, but not 2.5 μM (*Figure 3B*). The effect of BMS-345541 was confirmed in B16F10 (*Figure 3C*), CFPAC-1 (*Figure 3D*), and HCT116 (*Figure 3E*) cells. Furthermore, BQ treatment (24 hr) of HCT116 cells caused increased cell surface expression of MHC-I, which could

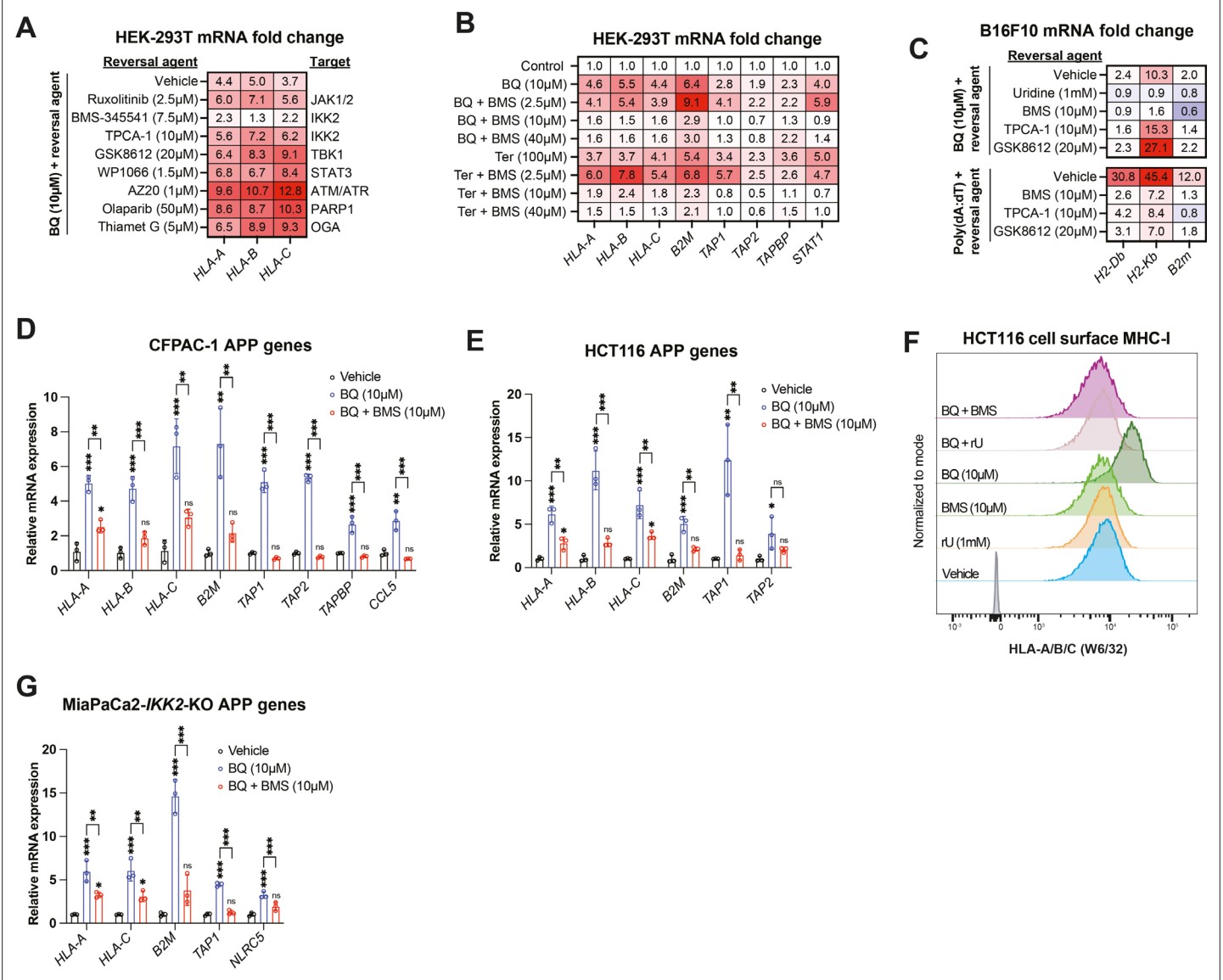

**Figure 3.** IKK2 inhibitor BMS-345541 abrogates brequinar (BQ)-mediated APP induction in an IKK2-independent manner. (**A, B**) HEK-293T cells were treated with indicated agents for 24 hr and then subjected to RT-qPCR quantification of mRNA levels for indicated APP genes. Numbers in the heatmap represent mean of four determinations. (**C–E, G**) B16F10 (**C**), CFPAC-1 (**D**), HCT116 (**E**), or MiaPaCa2-IKK2-KO (**G**) cells were treated with indicated agents for 24 hr and subjected to RT-qPCR analysis of indicated genes. Data in (**D, E, G**) represent mean ± SD of three independent experiments. *p<0.05, **p<0.01, and ***p<0.001 by one-way ANOVA with Tukey's multiple-comparison test. For (**C**), numbers in the heatmap represent mean fold change versus vehicle with three determinations; representative results for three independent experiments are shown. (**F**) Flow cytometry analysis of cell surface MHC-I in live HCT116 cells treated with indicated agents for 24 hr.

The online version of this article includes the following source data and figure supplement(s) for figure 3:

**Source data 1.** Source data for RT-qPCR experiments shown in *Figure 3* and *Figure 3—figure supplement 1*.

**Figure supplement 1.** IKK2, JAK1, JAK2, and TBK1 are dispensable for BQ-mediated APP induction.

be reversed by either uridine supplementation or by treatment with BMS-345541; neither uridine nor BMS-345541 alone affected cell surface MHC-I expression (*Figure 3F*).

Given that TPCA-1 (an established IKK2 inhibitor; *Podolin et al., 2005*) did not block BQ-mediated APP induction (*Figure 3A and C*), we suspected that this effect of BMS-345541 was independent of IKK2. To test this, we used previously reported MiaPaCa2 cells with CRISPR-Cas9 deletion of *IKK2* (MiaPaCa2-*IKK2*-KO) (*Napoleon et al., 2022*). Increased APP mRNA expression was observed upon BQ, teriflunomide, or GSK983 treatment (all DHODH inhibitors) of either wild-type or *IKK2*-KO

MiaPaCa2 cells (*Figure 3—figure supplement 1C*). However, while TNF-alpha stimulation induced APP and *CCL5* a canonical NF-κB target gene downstream of TNF-alpha (*Yeo et al., 2020*) expression in wild-type cells, this was not observed in *IKK2*-KO cells, as expected (*Figure 3—figure supplement 1C*, far right). Finally, BQ-mediated APP induction in *IKK2*-KO cells was significantly reversed with concurrent BMS-345541 treatment (*Figure 3G*). Together, these results demonstrate that IKK2 is dispensable for BQ-mediated APP induction and that the observed reversal effect of BMS-345541 is independent of IKK2.

## Nucleotide starvation induces APP transcription in a P-TEFb-dependent manner

To further investigate the mechanism by which BMS-345541 blocks APP induction downstream of pyrimidine starvation, we leveraged publicly available data on the target profile of BMS-345541 and other agents tested in the cell-free KINOMEscan assay (*Davis et al., 2011*). BMS-345541 reproducibly bound more than 20 kinases, with dissociation constants ($k_d$) ranging from 130 to 8100 nM (*Figure 4A*). We prioritized potential targets with a $k_d$ in the low micromolar range, given that 2.5 µM BMS-345541 did not block BQ-mediated APP induction in our previous experiments, and the effect seemed to be maximal at 10 µM, with no significant increase in the magnitude of the effect between 10 µM and 40 µM (*Figure 3B*). Additionally, we prioritized targets that were >50% inhibited with 10 µM BMS-345541 treatment. These two conditions correspond to the upper left quadrant of *Figure 4A*.

One potential target that met the selection criteria was CDK9, which together with cyclin T1 or T2 forms positive transcription elongation factor B (P-TEFb). P-TEFb is required for the release of promoter-proximal paused RNA polymerase II (Pol II) into productive elongation and therefore is essential for Pol II transcription from paused promoters (*Price, 2000*; *Ni et al., 2008*). The potent P-TEFb inhibitor flavopiridol (*Chao et al., 2000*) phenocopied BMS-345541 in our assays as it blocked APP induction downstream of DHODH, IMPDH1/2 (by mycophenolate), or CTP synthase (by 3-deazauridine *McPartland et al., 1974*) inhibition (*Figure 4—figure supplement 1A*). This suggests that APP induction downstream of nucleotide starvation requires P-TEFb-mediated paused Pol II release. It also suggests that the BMS-345541 effect of reversing BQ-induced APP upregulation is due to P-TEFb inhibition.

Within the list of kinases bound by BMS-345541 (*Figure 4A*), we eliminated those that were (a) not expressed by CFPAC-1 cells in our RNA-seq data, (b) not bound by flavopiridol in KINOMEscan data, or (c) bound by ruxolitinib in KINOMEscan data with Kd < 500 nM (as 2.5 µM ruxolitinib failed to reverse BQ-mediated APP induction; *Figure 3A*). Five candidates (besides CDK9) remained that were bound by both BMS-345541 and flavopiridol in KINOMEscan assays. Of these, three are CDKs known to play a role in transcription (CDK7, CDK13, and CDK16). Inhibition of any of these CDKs could theoretically account for the observed effects of flavopiridol and BMS-345541. However, previous studies suggest that flavopiridol inhibition of these CDKs in vivo is much less efficient than in cell-free assays because it is competitive with ATP (and thus less efficient in living cells where the ATP concentration is in the 1–10 mM range, which is much higher than in cell-free assay conditions), while its inhibition of P-TEFb is not affected by ATP concentration (*Chao et al., 2000*). Furthermore, flavopiridol and the CDK7 inhibitor THZ1 have very different (and mutually exclusive) effects on transcriptional processes (*Nilson et al., 2015*), arguing against CDK7 inhibition as the mechanism of flavopiridol's effect.

To further probe whether the observed effect of flavopiridol was due to CDK9 inhibition, we tested two other CDK9 inhibitors (AT7519 and dinaciclib). Both CDK9 inhibitors phenocopied flavopiridol in our assays (*Figure 4B*). Furthermore, a previously characterized CDK9-targeted proteolysis targeting chimera (PROTAC), termed PROTAC2 (*King et al., 2021*), had the same effect (*Figure 4B*). PROTAC2 consists of a CDK9-binding aminopyrazole warhead conjugated to pomalidomide, which recruits the E3 ubiquitin ligase Cereblon (*CRBN*). Cereblon in turn ubiquitinates CDK9, resulting in its proteasomal degradation. Co-treatment of HEK-293 cells with PROTAC2 and pomalidomide prevents PROTAC2-mediated CDK9 degradation, as free pomalidomide competes with PROTAC2 for Cereblon binding (*King et al., 2021*). We observed that PROTAC2 (1 µM) blocked BQ-mediated APP induction, and this effect was reversed by co-treatment with tenfold excess pomalidomide (10 µM); however, when we increased the concentration of PROTAC2 to 10 µM (so that PROTAC2 and pomalidomide concentrations were equal), pomalidomide no longer had this effect (*Figure 4C*). Consistently, immunoblot analysis showed that 10 µM pomalidomide prevents CDK9 degradation upon 1 µM but not

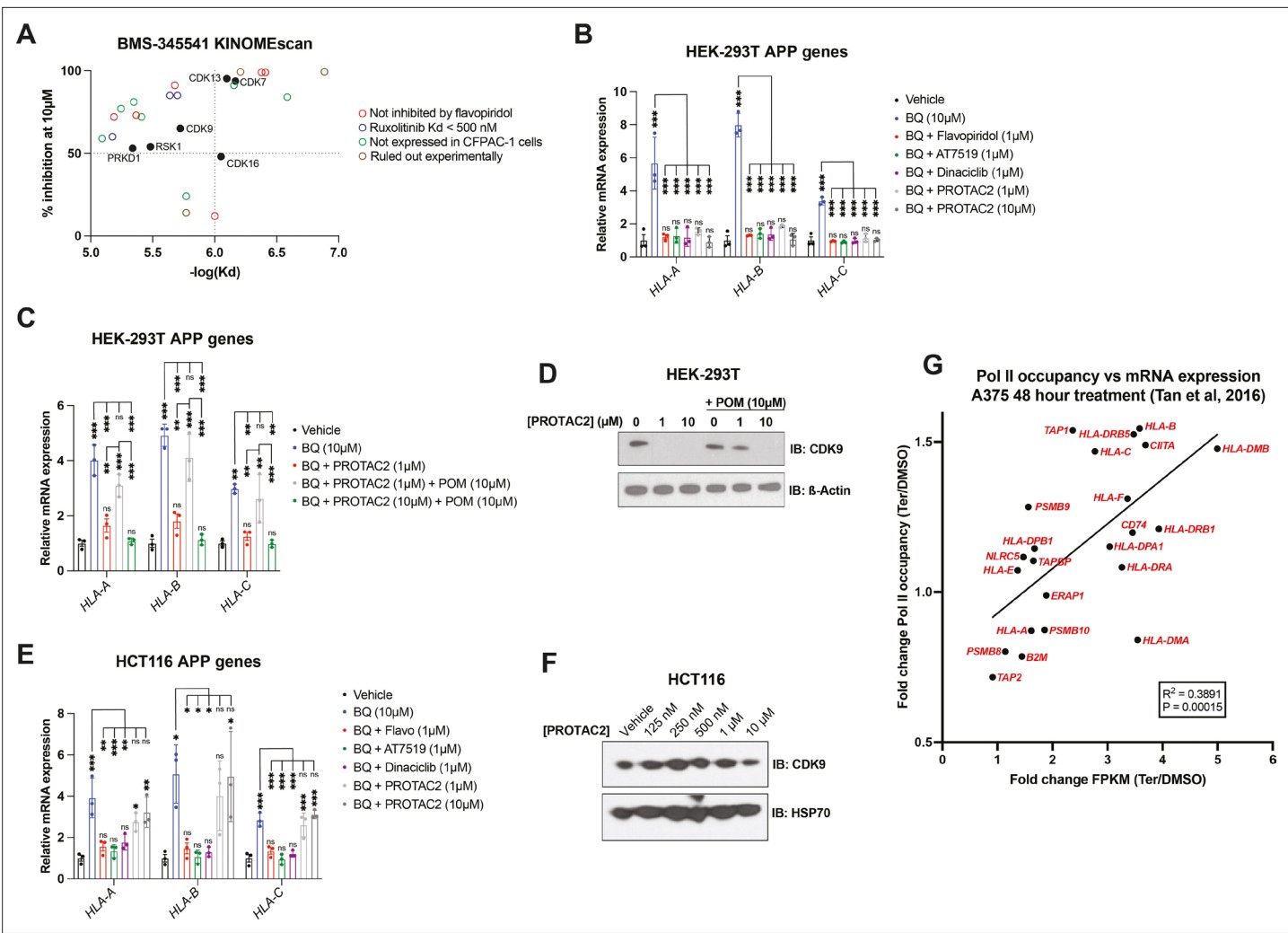

**Figure 4.** P-TEFb inhibitor flavopiridol abrogates APP induction downstream of nucleotide depletion. (**A**) Plot of percent inhibition (10 μM treatment) vs. -log(dissociation constant) for kinases bound by BMS-345541 in cell-free KINOMEscan assays; data derived from ***Davis et al., 2011***. Each data point represents an individual kinase. (**B, C**) RT-qPCR quantification of mRNA levels for indicated APP genes in HEK-293T cells treated with indicated agents for 24 hr. Data represent mean ± SD of three independent experiments. **p<0.01, ***p<0.001 and 'ns' p>0.05 by one-way ANOVA with Tukey's multiple-comparison test (**C**). (**D**) Western blot analysis for CDK9 performed on HEK-293T cells treated with CDK9-targeted PROTAC (PROTAC2) and/or pomalidomide (POM) for 24 hr. Beta actin was used as a loading control. (**E**) RT-qPCR quantification of mRNA levels for indicated APP genes after 24 hr treatment with indicated agents. Data represent mean ± SD of three independent experiments. *p<0.05, **p<0.01, and ***p<0.001 by one-way ANOVA Tukey's multiple-comparison test. (**F**) Western blot analysis for CDK9 performed on HCT116 cells treated with the indicated concentrations of PROTAC2 for 24 hr. Heat shock protein 70 (HSP70) was used as a loading control. (**G**) Linear regression analysis of fold change (teriflunomide/DMSO) in Pol II occupancy (assessed by ChIP-seq) vs. fold change (teriflunomide/DMSO) in mRNA abundance (assessed by RNAseq) following 48 hr treatment of A375 cells with teriflunomide (25 μM) or DMSO vehicle control; data derived from ***Tan et al., 2016***.

The online version of this article includes the following source data and figure supplement(s) for figure 4:

**Source data 1.** Source data for RT-qPCR experiments and KINOMEscan data analysis shown in *Figure 4* and *Figure 4—figure supplement 1*.

**Source data 2.** Source data for western blot images shown in *Figure 4*.

**Figure supplement 1.** Flavopiridol reverses MHC-I transcriptional induction downstream of nucleotide depletion.

10 μM PROTAC2 treatment (*Figure 4D*). When we repeated the experiment shown in *Figure 4B* with HCT116 cells, we found that all CKD9 inhibitors reversed BQ-mediated APP induction, but PROTAC2 did not (*Figure 4E*). Concordantly, immunoblot analysis showed that PROTAC2 did not cause CDK9 depletion in HCT116 cells treated in parallel (*Figure 4F*). Taken together, these results demonstrate that CDK9 degradation is necessary for the reversal effect of PROTAC2 and that CDK9 is required for BQ-mediated APP induction.

The dependence of BQ-mediated APP induction on CDK9 strongly suggests that nucleotide starvation enforces nascent transcription of APP genes, as opposed to increased mRNA stability. This is further supported by the rapid buildup of APP transcripts following DHODH inhibitor treatment (within 4 hr, *Figure 3—figure supplement 1A*). Additionally, ChIP-seq analysis of global Pol II occupancy following 48 hr teriflunomide treatment in A375 cells (*Tan et al., 2016*) shows increased Pol II occupancy across many APP genes, and fold change in Pol II occupancy significantly correlated with fold change in mRNA expression under the same conditions (*Figure 4G*). Overall, these results show that nucleotide starvation induces an antigen presentation gene expression program that is independent of canonical APP regulators but depends on CDK9/P-TEFb.

## BQ suppresses tumor growth, induces MHC-I expression, and increases immunotherapy efficacy in a syngeneic melanoma model

Enforced MHC-I upregulation by various interventions can facilitate anticancer immunity and enhance the efficacy of ICB by antibodies directed at PD-(L)1 and/or CTLA-4 (*Yamamoto et al., 2020*; *Goel et al., 2017*; *Kalbasi et al., 2020*; *Gu et al., 2021*). Moreover, high MHC-I expression has been proposed as a predictor of ICB response (*Rodig et al., 2018*; *Liu et al., 2019*; *Grasso et al., 2020*; *Shklovskaya et al., 2020*), and high expression of MHC-I and other APP genes, including *NLRC5* and *TAP1*, correlates with better survival in patients with melanoma (*Figure 5—figure supplement 1A*), for whom ICB is a first-line therapy. Therefore, we asked if BQ could improve anticancer immunity in the B16F10 melanoma immunocompetent mouse model, which is typically refractory to dual ICB (i.e., anti-PD-1 plus anti-CTLA-4) (*Twyman-Saint Victor et al., 2015*).

BQ (10 mg/kg daily IP injection) markedly suppressed tumor growth and led to reduced tumor burden (*Figure 5A and B*). Historically, the lead tool compound that was ultimately modified to BQ (called NSC 339768) was prioritized in part based on its activity against B16 melanoma *Dibner et al., 1985*; however, to our knowledge, this is the first direct demonstration of BQ activity in this model system. Consistent with our in vitro metabolomics data (*Figure 1I and J*, *Figure 1—figure supplement 1D and E*), BQ treatment caused marked buildup of metabolites upstream of DHODH and depletion of downstream pyrimidine nucleotide species in B16F10 tumors (*Figure 5C*), confirming target engagement in vivo. Metabolomics analysis of BQ- and vehicle-treated tumors separated in principal component analysis (*Figure 5—figure supplement 1B*) and unsupervised hierarchical clustering (*Figure 5—figure supplement 1C*), confirming the perturbation of tumor metabolism following DHODH inhibition.

BQ-treated B16F10 tumors showed increased mRNA expression of MHC-I (*H2-Db* and *H2-Kb*) and *Nlrc5* (*Figure 5D*). We thus addressed whether BQ could augment the efficacy of dual ICB (anti-CTLA-4 plus anti-PD-1) with the knowledge that enforced MHC-I antigen presentation has also been shown to boost the effect of ICB (*Yamamoto et al., 2020*; *Kalbasi et al., 2020*; *Gu et al., 2021*). While BQ is not an approved medication, two FDA-approved low-potency DHODH inhibitors (leflunomide, teriflunomide) are effective in treating autoimmune conditions such as rheumatoid arthritis and multiple sclerosis and act to decrease the activity of auto-reactive T-lymphocytes (*Klotz et al., 2019*; *Fox et al., 1999*; *Miller, 2021*). It was possible that BQ treatment may actually impair the effectiveness of ICB by inhibiting T-lymphocytes despite augmented cancer cell antigen presentation. We, therefore, tested both concurrent, upfront administration of BQ plus dual ICB and sequential administration of BQ followed by dual ICB (*Figure 5—figure supplement 1D*).

Similar to its impressive activity in our first experiment (*Figure 5A and B*), BQ monotherapy conferred marked survival benefit. This was significantly enhanced by subsequent dual ICB, while dual ICB alone conferred only marginally prolonged survival, and concurrent BQ plus dual ICB did not significantly improve survival versus BQ monotherapy (*Figure 5E*). This suggests that sequential (rather than concurrent) administration of DHODH inhibitor and ICB may be superior. Hypotheses that may explain these findings include: (a) concurrent BQ dampens the initial anticancer immune response generated by dual ICB, or (b) cancer cell MHC-I and related genes are not maximally upregulated at the time of ICB administration with concurrent treatment. Taken together, these results show that BQ causes pyrimidine nucleotide depletion, MHC-I and APP gene transcriptional upregulation, and additive survival benefit with dual ICB in a highly aggressive and ICB-refractory mouse melanoma model.

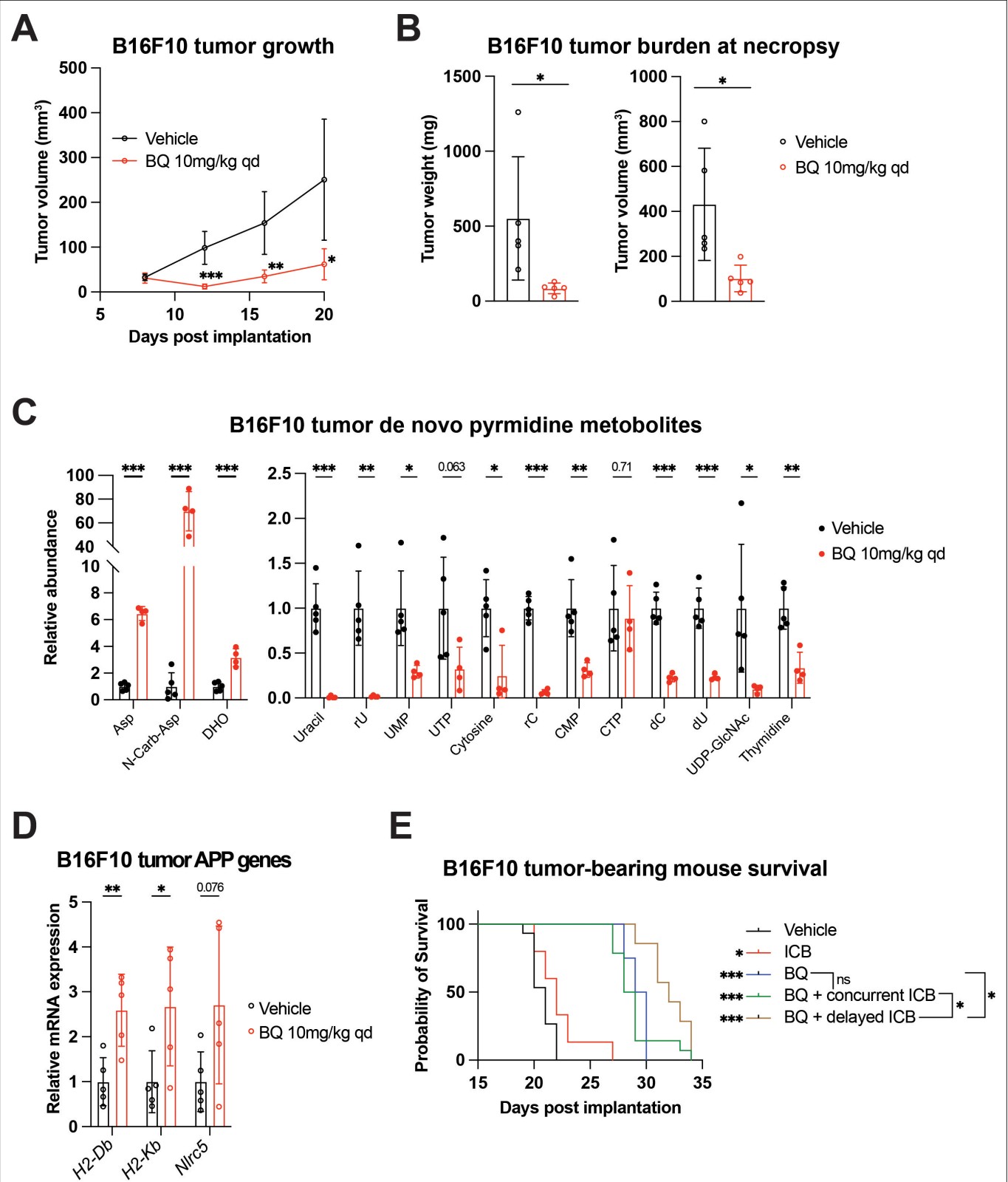

**Figure 5.** Brequinar (BQ) inhibits tumor growth, increases tumor MHC-I, and enhances immune checkpoint blockade efficacy in B16F10 murine melanoma model. (**A–D**) B16F10 cells were injected subcutaneously into syngeneic C57Bl/6J hosts. Tumor-bearing mice were treated with BQ (10 mg/kg, administered intraperitoneally daily) or vehicle control starting at day 7 post implantation. (**A**) Longitudinal estimation of tumor volume using digital caliper measurement of B16F10 subcutaneous tumors in BQ-treated and vehicle-treated tumor-bearing mice. Data represent mean ± SD of n = 5 mice

*Figure 5 continued on next page*

*Figure 5 continued*

per group. *p<0.05, **p<0.01, ***p<0.001 by unpaired *t*-tests with Benjamini and Hochberg FDR correction. (**B**) Weight (left) and volume (right) of tumors at necropsy. Data represent mean ± SD of n = 5 mice per group. *p<0.05 by unpaired *t*-test. (**C**) Quantification of indicated metabolites from B16F10 tumors harvested at necropsy. Data represent mean ± SD of n = 5 mice in control group and n = 4 for BQ group; one sample was excluded due to sample attrition during processing, leading to insufficient metabolite recovery. *p<0.05, **p<0.01, ***p<0.001 by unpaired *t*-test. (**D**) RT-qPCR quantification of mRNA expression for indicated APP genes performed on tumors harvested at necropsy. Data represent mean ± SD of n = 5 mice per group. * p<0.05 and **p<0.01 by unpaired *t*-test. (**E**) Kaplan–Meier survival analysis for mice implanted with B16F10 tumors as in (**A–D**) and treated with indicated regimens; see *Figure 5—figure supplement 1D* for treatment timeline. *p<0.05, ***p<0.001 by Mantel–Cox logrank test. Sample size (n): vehicle (black), n = 15; immune checkpoint blockade (ICB; Anti-CTLA-4 and anti-PD-1; 100 μg/mouse each, IP twice per week) (red), n = 15; BQ + concurrent ICB (green), n = 14; BQ monotherapy (blue), n = 7; BQ + delayed ICB (brown), n = 8.

The online version of this article includes the following source data and figure supplement(s) for figure 5:

**Source data 1.** Source data for tumor burden, tumor metabolomics, tumor RT-qPCR, and mouse survival experiments shown in *Figure 5* and *Figure 5—figure supplement 1*.

**Figure supplement 1.** Increased mRNA expression of APP genes correlates with longer survival of patients with melanoma.

## Discussion

Our results demonstrate that pyrimidine nucleotide depletion by DHODH inhibition causes increased expression of APP genes and increased antigen presentation via MHC-I across a diverse panel of cancer cell lines (*Figure 1*). This effect of BQ and teriflunomide is strictly dependent on pyrimidine nucleotide depletion, as it was abrogated by restoration of pyrimidine levels with exogenous uridine (*Figure 2B–D*, *Figure 2—figure supplement 1B–E*). Furthermore, genetic deletion of *DHODH* recapitulated this effect, and treatment of DHODH knockout cells with BQ did not further increase MHC-I mRNA expression (*Figure 2E*). Our inhibitor reversal studies determined that BQ-mediated APP induction is independent of several canonical APP regulatory pathways, including IFN-JAK-STAT, cGAS-STING-TBK1, and NF-κB (*Figure 3*, *Figure 3—figure supplement 1*). We showed that this effect relies on P-TEFb-mediated release of Pol II from promoter-proximal paused state to productive elongation (*Figure 4*). These findings were extended to inhibition of IMPDH (which depletes cellular GTP) and CTPS (which depletes cellular CTP), as these effects were also reversible with P-TEFb inhibition (*Figure 4—figure supplement 1A*). This suggests that pharmacologic depletion of these nucleotide species also triggers APP upregulation in a P-TEFb-dependent manner.

Since T-cell recognition of antigens via MHC-I is required for T-cell-mediated elimination of cancer cells or virus-infected cells, these results have important implications for the development of nucleotide synthesis inhibitors as anticancer/antiviral therapies. We provide proof-of-concept evidence that pretreatment with DHODH inhibitors can improve the efficacy of ICB in a highly aggressive and ICB-refractory mouse melanoma model (*Figure 5*, *Figure 5—figure supplement 1*). Because BQ-mediated APP induction does not require interferon signaling, this strategy may have particular relevance for clinical scenarios in which tumor antigen presentation is dampened by the loss or silencing of cancer cell interferon signaling, which has been demonstrated to confer both intrinsic (*Shin et al., 2017*) and acquired (*Zaretsky et al., 2016*) ICB resistance in human melanoma patients.

Emerging evidence suggests that cancer cell MHC-I expression predicts favorable response to ICB, and several recent studies have shown that enforced cancer cell MHC-I expression enhances anti-cancer immunity and ICB efficacy in various mouse models. Accordingly, functional genomic screens for regulators of cancer cell MHC-I expression have recently been undertaken, and these efforts have revealed novel molecular targets to induce cancer cell APP activity (*Gu et al., 2021*; *Dersh et al., 2021*). Agents shown to increase cancer cell antigen presentation include hydroxychloroquine (by autophagy inhibition) (*Yamamoto et al., 2020*), poly(I:C) (by NF-κB activation downstream of dsRNA sensing) (*Kalbasi et al., 2020*), SMAC mimetics (by NF-κB activation) (*Gu et al., 2021*), CDK4/6 inhibitors (by activation of endogenous genomic retroviral elements) (*Goel et al., 2017*), and others. It is very likely that many other anticancer drugs perturb cancer cell antigen presentation and/or have other immunomodulatory properties in addition to their cell-intrinsic antiproliferative activity (*Petroni et al., 2021*), and this area requires further scrutiny. In this study, we identified DHODH inhibition as a powerful inducer of antigen presentation and MHC-I expression in diverse cancer cell lines and in HEK-293T cells.

Previous studies have linked pyrimidine depletion with upregulation of innate immunity and interferon-stimulated genes (*Lucas-Hourani et al., 2013*; *Sprenger et al., 2021*) and this was

confirmed by our transcriptomic profiling experiments (*Figure 1B and C*). Multiple mechanistic explanations for these observations have been suggested. Lucas-Hourani et al. proposed that interferon-stimulated gene expression requires the DNA damage checkpoint kinase ATM (*Lucas-Hourani et al., 2013*) while Sprenger et al. conclude that pyrimidine depletion causes accumulation of mitochondrial DNA in the cytosol, which is sensed by the cGAS-STING-TBK1 pathway to promote innate immunity (*Sprenger et al., 2021*). In our models, neither ATM/ATR nor TBK1 inhibition blocked BQ-mediated APP induction (*Figure 4A*). It is possible that pyrimidine nucleotide shortage leads to APP induction by multiple redundant mechanisms, any of which may predominate based on the cellular context. We speculate that cells may have evolved multiple means of sensing acute pyrimidine shortage as a way to detect viral infection or malignant transformation, as both viral replication and uncontrolled cell proliferation avidly consume nucleotides.

Our focused chemical screen for MHC-I inducers (*Figure 2F*) identified the approved IMPDH1/2 inhibitor mycophenolate, which was subsequently validated in multiple other cell types (*Figure 2— figure supplement 1D*, S4A). This is consistent with a recent study in which IMPDH inhibition was shown to enhance ICB efficacy by favorably altering the MHC-I peptide repertoire and increasing immunoproteasome expression (*Keshet et al., 2020*). However, in this study, the cancer cells were pretreated with IMPDH inhibitor before implantation into syngeneic hosts, and so possible counter-vailing immunosuppression by systemic IMPDH inhibitor treatment was not addressed (*Keshet et al., 2020*). Our in vivo results (*Figure 5E*) highlight the importance of timing/sequence when administering immunotherapy in combination with nucleotide synthesis inhibitors and suggest that upfront BQ followed by ICB may be superior to concurrent administration.

Thymidylate synthase inhibition was recently shown to induce MHC-I in a model of diffuse large B cell lymphoma (*Dersh et al., 2021*). The failure of thymidylate synthase inhibitors 5-fluorouracil and methotrexate to induce MHC-I in our screen (*Figure 2F*) may be attributable to cell line differences (PDAC vs. DLBCL), dose/duration considerations, or the use of different thymidylate synthase inhibitors than in their study (which used pemetrexed and raltitrexed). Thus, it appears that the abundance of multiple nucleotide species can exert context-dependent influence on MHC and APP gene expression, and key details of this relationship remain to be elucidated.

Overall, our study establishes P-TEFb and Pol II elongation control as a mechanistic link between nucleotide depletion and APP induction. We provide proof-of-concept evidence for combinatorial benefit of DHODH inhibition and ICB in an aggressive and poorly immunogenic mouse model of melanoma. A deeper understanding of metabolic control of antigen presentation will enable rational therapy development for cancer and viral infection.

# Materials and methods

**Key resources table**

| Reagent type (species) or resource | Designation | Source or reference | Identifiers | Additional information |
|---|---|---|---|---|
| Strain (*Mus musculus*) | C57BL/6J | JAX | RRID:IMSR_JAX:000664 | Mouse strain used for tumor implantation experiments |
| Cell line (*Homo sapiens*) | A375 | ATCC | CRL-1619 | Source: malignant melanoma, 54-year-old female |
| Cell line (*H. sapiens*) | A549 | ATCC | CCL-185 | Source: lung carcinoma, 58-year-old male |
| Cell line (*H. sapiens*) | CFPAC-1 | ATCC | CRL-1918 | Source: pancreas adenocarcinoma, 26-year-old male |
| Cell line (*H. sapiens*) | HCT116 | ATCC | CCL-247 | Source: colorectal carcinoma, adult male (age unspecified) |
| Cell line (*H. sapiens*) | HEK-293T | ATCC | CRL-3216 | Source: kidney, female embryo |
| Cell line (*H. sapiens*) | MDA-MB-231 | ATCC | HTB-26 | Source: breast adenocarcinoma, 51-year-old female |

*Continued on next page*

*Continued*

| Reagent type (species) or resource | Designation | Source or reference | Identifiers | Additional information |
|---|---|---|---|---|
| Cell line (*H. sapiens*) | Panc1 | ATCC | CRL-1469 | Source: pancreas adenocarcinoma, 56-year-old male |
| Cell line (*H. sapiens*) | MiaPaCa2 (wild-type parental cell line for MiaPaCa2-IKK2-KO) | Gift from Amar Natarajan laboratory; **Napoleon et al., 2022**, originally from ATCC | CRL-1420 | Source: pancreas adenocarcinoma, 65-year-old male |
| Cell line (*H. sapiens*) | MiaPaCa2-IKK2-KO | Gift from Amar Natarajan laboratory; **Napoleon et al., 2022** | | Please see **Napoleon et al., 2022** for information on how the cell line was generated |
| Cell line (*H. sapiens*) | S2-013 | Tony Hollingsworth laboratory; **Mullen et al., 2023** | RRID:CVCL_B280 | Source: liver metastasis from pancreas carcinoma, 73-year-old male |
| Cell line (*H. sapiens*) | S2-013 sgNT | **Mullen et al., 2023** | | S2-013 stably transduced with non-targeting sgRNA vector and Cas9 |
| Cell line (*H. sapiens*) | S2-013 sgDHODH | **Mullen et al., 2023** | | S2-013 stably transduced with DHODH-targeting sgRNA vector and Cas9 |
| Cell line (*M. musculus*) | B16F10 | ATCC | CRL-6475 | Source: malignant melanoma, male C57BL/6 mouse |
| Antibody | Anti-HSP70 (host: rabbit polyclonal) | CST | Ca# 4872 | Dilution factor 1:1000 for western blot |
| Antibody | Anti-CDK9 (host: rabbit monoclonal) | CST | Cat# 2316 Clone: C12F7 | Dilution factor 1:1000 for western blot |
| Antibody | Anti-ACTB (host: mouse monoclonal) | Santa Cruz Biotechnology | Cat# sc-4778 Clone: C4 | Dilution factor 1:500 for western blot |
| Antibody | Anti-H2-Db (host: mouse monoclonal) | BioLegend | Cat# 111508 Clone: HK95 | Conjugated to phycoerythrin for flow cytometry Dilution factor 2 µl in 100 µl final volume |
| Antibody | Anti-MHC-I (host: mouse monoclonal) | BioLegend | Cat# 311418 Clone: W6/32 | Conjugated to Pacific Blue fluorescent marker for flow cytometry Dilution factor 2 µl in 100 µl final volume |
| Antibody | Anti-MHC-I (host: mouse monoclonal) | BioLegend | Cat# 311406 | Conjugated to phycoerythrin for flow cytometry Dilution factor 2 µl in 100 µl final volume |
| Chemical compound, drug | Please see **Supplementary file 1** for complete list of all biologically active chemical compounds used in this study | Please see **Supplementary file 1** for complete list of all biologically active chemical compounds used in this study | Please see **Supplementary file 1** for complete list of all biologically active chemical compounds used in this study | Please see **Supplementary file 1** for complete list of all biologically active chemical compounds used in this study |
| Gene (*H. sapiens*) | Please see **Supplementary file 2** for complete list of all genes mentioned in this study | Please see **Supplementary file 2** for complete list of all genes mentioned in this study | Please see **Supplementary file 2** for complete list of all genes mentioned in this study | Please see **Supplementary file 2** for complete list of all genes mentioned in this study |
| Gene (*M. musculus*) | Please see **Supplementary file 2** for complete list of all genes mentioned in this study | | Please see **Supplementary file 2** for complete list of all genes mentioned in this study | Please see **Supplementary file 2** for complete list of all genes mentioned in this study |
| Sequence-based reagent: RT-qPCR primer sets (*H. sapiens*) | Please see **Supplementary file 2** for complete list of all primer sets used in this study | Please see **Supplementary file 2** for complete list of all genes mentioned in this study | Please see **Supplementary file 2** for complete list of all primer sets used in this study | Please see **Supplementary file 2** for complete list of all primer sets used in this study |

## Cell culture and cell lines

The S2-013 cell line is a clonal derivative of the Suit2 cell line and was a kind gift from the Tony Hollingsworth laboratory at the University of Nebraska Medical Center. The MiaPaCa2 *IKK2*-KO and parental wild-type MiaPaCa2 cell lines were a kind gift from the Amar Natarajan laboratory at the University of Nebraska Medical Center. All other cell lines in this study were obtained from American Type Culture Collection (Manassas, VA). All human cell lines were authenticated by STR profiling by the Genetics Core at the University of Arizona. Cells were routinely (at the time of initial revival from liquid nitrogen storage and at least every 6 months) determined to be free of mycoplasma contamination by PCR-based methods. Cells were cultured in Dulbecco's modified Eagle medium (Sigma-Aldrich, St Louis, MO) supplemented with 50 IU/mL penicillin, 50 µg/mL streptomycin, and incubated at 37°C in a humidified incubator with 5% $CO_2$. Cells were maintained at 10% FBS. Upon reaching 70–80% confluency, cells were passaged by washing with phosphate-buffered saline (PBS) before adding 0.25% trypsin (Caisson Labs, Smithfield, UT) and plating at 25% confluency.

## Drug treatment of cultured cells for RT-qPCR and flow cytometry experiments

Drug treatment dose and duration are indicated for each experiment. A table of manufacturer and catalog number for each agent described can be found in *Supplementary file 1*. For stimulation with poly(dA:dT), 2 µg of poly(dA:dT) and 2 µL of Lipofectamine2000 (Invitrogen #11668027) were incubated in 400 µL Opti-MEM (Gibco #11058021) for 30 min at room temperature and then added to cells in 2 mL final volume of complete media.

## Cell viability assays

Cells were seeded in 96-well plates (1000 cells per well in 90 µL media) and allowed to equilibrate overnight. Cells were then treated with indicated compounds (final volume 100 µL) for 72 hr, and viability was assessed by CellTiter-Glo assay (Promega, Madison, WI). Luminescence values for each condition were normalized to the average luminescence of the vehicle-treated control replicates. Experiments were performed at least three times using biological triplicates for each condition. Dose–response curves were fit to a nonlinear regression model using Prism9 software.

## Liquid chromatography–tandem mass spectrometry-based metabolomics analysis

For in vitro metabolomics experiments, $5 \times 10^5$ cells were seeded in 6-well plates and allowed to equilibrate overnight. At the start of each assay, the cell culture media was changed, and fresh media with desired conditions was added (to eliminate metabolite depletion from overnight equilibration as a confounding variable). Following 8 hr treatment of cancer cell lines with BQ (or in the case of *Figure 2B*, 24-hr treatment with BQ ± 1 mM uridine), polar metabolites were extracted and quantified as previously described (*Olou et al., 2020*). For B16F10 tumor metabolomics, subcutaneous tumors were harvested at necropsy and immediately snap frozen in liquid nitrogen and stored at –80°C. Tumors were subsequently ground into fine powder in liquid nitrogen using a mortar and pestle, and metabolites were extracted using the same method as for cultured cells. Peak areas were normalized to the mass of tumor tissue that was input.

Datasets were processed using Skyline (MacCoss Lab Software), and Metaboanalyst5.0 web tool was used to generate principal component analysis and heatmap visualizations of resulting datasets. Relative metabolite abundances were normalized to the average peak area of the experimental control group.

## Mice studies

All procedures were approved by the Institutional Animal Care and Use Committee (IACUC) at the University of Nebraska Medical Center (protocol number: 20-112-03-FC). For tumor xenograft studies, $10^4$ B16F10 cells in a 1:1 vol/vol ratio (100 µL final volume) with Matrigel were injected subcutaneously into the right flank of 10-week-old female C57BL/6J mice (Jackson Labs). Tumors of live mice were serially measured in two dimensions using digital calipers, and tumor volume for *Figure 5A* was calculated as $(0.5 \text{ L} \times \text{W}^2)$, where L is the longest measurable tumor dimension and W is the longest tumor dimension that is perpendicular to L. For *Figure 5B and C*, tumors were harvested at necropsy,

weighed on an analytical balance (for *Figure 5B*), and measured in three perpendicular dimensions by calipers to generate volume measurements for *Figure 5C*, which were calculated as (dimension 1 × dimension 2 × dimension 3).

For survival experiments (*Figure 5E*), mice were monitored daily for signs of euthanasia criteria or actual demise. When tumor volume reached 2000 mm$^3$ as determined by the above formula for live mice (0.5 L × W$^2$), mice were sacrificed according to protocol euthanasia criteria.

BQ was obtained from Clear Creek Bio and dissolved in 0.9% NaCl. For both endpoint and survival studies, BQ (10 mg/kg) or vehicle solvent (0.9% NaCl) was injected intraperitoneally daily. Anti-CTLA-4 and anti-PD-1 antibodies, as well as their respective isotype controls, were obtained from BioXCell. Antibodies were dosed at 100 µg/mouse IP twice per week. See *Figure 5—figure supplement 1D* for treatment regimen timeline.

## RNA sequencing and gene set enrichment analysis

For RNA sequencing experiments, S2-013 or CFPAC-1 cells were treated with BQ for the indicated dose and duration (*Figure 1* and S1). For 2-week drug treatment experiments, cells were passaged every 3 days and 5 × 10$^5$ cells were reseeded in a new 10 cm tissue culture dish. RNA was isolated using RNEasy Mini kit (QIAGEN, Cat# 74104).

Samples were processed by BGI Genomics (San Jose, CA) according to their proprietary method. Briefly, RNA quality check was performed using Agilent 2100 Bioanalyzer. Poly-A-containing mRNA was isolated using magnetic beads and then fragmented using divalent cations under elevated temperature. cDNA synthesis was performed using reverse transcriptase and RNase H. Adapter sequences were then ligated onto cDNA fragments, purified, enriched by PCR, quantified by Qubit, and pooled to generate the final library. Libraries were then sequenced using the BGI DNBseq platform. Reads mapped to rRNA, low-quality reads, and reads with adaptors were removed. The resulting clean reads were mapped to the reference genome (hg19_UCSC_20180115) using HISAT2 program (http://www.ccb.jhu.edu/software/hisat/index.shtml) and converted to fragments per kilobase per million mapped reads (FPKM).

Fold change FPKM (BQ/vehicle control) values for all expressed genes were subjected to gene set enrichment analysis (*Subramanian et al., 2005*) with GSEA prerank using HALLMARK and KEGG genes sets from the Molecular Signatures Database (MSigDB) as previously described (*Dasgupta et al., 2020*). Gene sets positively enriched with FDR q < 0.25 are shown in *Figure 1B*.

## Real-time quantitative PCR analysis for mRNA expression

For in vitro RT-qPCR experiments, RNA was harvested using Trizol reagent (Thermo Fisher Scientific, Waltham, MA) according to manufacturer's instructions. For tumor RT-qPCR, tumors were crushed with mortar and pestle in liquid nitrogen, and Trizol was used to extract RNA from the resulting powder, just as for cultured cells. cDNA synthesis was performed (1 µg RNA input) using Bio-Rad (Hercules, CA) iScript cDNA synthesis kit (Cat# 1708891) according to manufacturer's instructions. For RT-qPCR reactions, 3 µL of diluted cDNA, 2 µL of primer mix (diluted to a final concentration of 200 nM for forward and reverse primers), and 5 µL SYBR green master mix (Thermo Fisher Cat# A25776) were mixed (10 µL final volume), and reactions were analyzed using Applied Biosystems QuantStudio5 instrument with previously reported thermocycling parameters (*Shukla et al., 2015*).

18S rRNA was used as a loading control to generate delta Ct values, and each sample was normalized to the experimental control delta Ct values to generate delta delta Ct values, which were converted to fold change by (fold change = 2^-ddCt). For all experiments, *ACTB* (beta-actin) mRNA expression was quantified and used as an additional loading control, and results were concordant regardless of whether 18S or *ACTB* was used for normalization. Primer sequences for RT-qPCR reactions are provided in *Supplementary file 2*.

## Flow cytometry measurement of cell surface MHC-I

Cells were treated as described and then detached with Accutase (Sigma Aldrich #A6964), washed twice with PBS, stained with fluorescent dye-conjugated antibodies against H2-Db (BioLegend #111508) or intact MHC-I, a heterodimer consisting of B2M and either HLA-A, HLA-B, or HLA-C (BioLegend #311418, BioLegend #311406) for 30 min at 4°C in PBS (2 µL antibody in final volume of 100 µL), washed once more with PBS, and then resuspended in FACS buffer and subjected to flow

cytometry analysis for fluorescence intensity. Aqua live/dead dye (Invitrogen #L34957) or propidium iodide was used to exclude dead cells from the analysis.

## Western blot

Protein isolation from cultured cells and western blotting procedure were described previously (*Olou et al., 2020*). CDK9 antibody was obtained from Cell Signaling Technology (#2316, clone C12F7), HSP70 antibody was obtained from Cell Signaling Technology (#4872), and beta-actin antibody was obtained from Santa Cruz Biotechnology (#sc-4778, clone C4). Blots were incubated with primary antibody overnight at 4°C in TBST with 5% milk protein, washed with TBST three times (5 min per wash), incubated with secondary antibody conjugated with horseradish peroxidase for 45 min at room temperature, again washed with TBST three times (5 min per wash), developed with ECL reagent, and visualized by autoradiography using plain film.

## Procurement and analysis of previously published datasets

All datasets reported by Tan and colleagues (*Tan et al., 2016*) were obtained from Gene Expression Omnibus, accession numbers GSE68053 and GSE68039. Processed RNA sequencing data for human A375 melanoma cells treated with DMSO vehicle control (GSM1661518, GSM1661518), or teriflunomide (25 µM) for 12 hr (GSM1661510, GSM1661511), 24 hr (GSM1661512, GSM1661513), 48 hr (GSM1661514, GSM1661515), or 72 hr (GSM1661516, GSM1661517) was downloaded as an Excel file from GSE68039 (GSE6809_A375.FPKM.xls) and directly analyzed by manual inspection. The two FPKM values for each experimental condition were averaged, and these average values were used to calculate the fold change (teriflunomide/DMSO) values presented in *Figures 1E* and *4G*.

For chromatin immunoprecitation sequencing (ChIP-seq) datasets (used to generate *Figure 4G*), Fastq files for human A375 melanoma cells treated for 48 hr with DMSO (GSM1661790) or teriflunomide (GSM1661791) were downloaded from GSE68053, trimmed of adapter sequences at the 3'ends with trim_galore v0.6 (https://github.com/FelixKrueger/TrimGalore; *Krueger, 2023*), and aligned to hg38 using Bowtie (v1.2.3) (*Langmead et al., 2009*; *Langmead and Rone, 2019*) with parameters `--minins 18 --maxins 1000 --fr --best --allow-contain`. Reads overlapping with the longest transcript of each gene (Genecode v32 and https://github.com/GeoffSCollins/PolTools/blob/master/PolTools/static/longest_transcript_with_downstream_start_codon.txt; *Collins, 2021*) were counted with BEDtools intersect (v2.27.1). Library size correction factors were calculated separately for the ChIP-seq datasets. The correction factor for a given ChIP-seq sample was computed by dividing the number of mapped reads in that sample by the average number of mapped reads across all ChIP samples (DMSO, A771726). After normalization, the total number of read counts (now corrected for total number of mapped reads per sample) aligned to each gene of interest were used to calculate fold change (teriflunomide/DMSO) in Pol II occupancy values presented in *Figure 4G*.

## Statistical analysis, hypothesis testing, and exclusion of data

For comparison of means between exactly two experimental groups, an unpaired *t*-test was used. For comparison of means between three or more experimental groups, a ANOVA was used. If the one-way ANOVA rejected the null hypothesis of all means being equivalent, a multiple comparison test was used to accept or reject the null hypothesis of equivalent means for each experimentally relevant pair-wise comparison. For each experiment, the chosen multiple comparison test set the family-wise type I error rate (i.e., alpha level) to 0.05 and computed multiplicity-adjusted p values for each pair-wise comparison. The choice of multiple comparison test was based on which pair-wise comparisons were of interest; this was prespecified during the design of each experiment. If each experimental group was to be compared to a single control group (e.g., for *Figure 1I and J*), Dunnett's multiple-comparison test was used, with each experimental group compared with the control group but not with the other non-control experimental groups. In all other cases, each experimental group (including the control group) was compared to every other experimental group (e.g., for *Figures 2B, C, D, E, 3D, E, G, 4B, C and E*) using Tukey's multiple-comparison test. Data were assumed to follow a Gaussian distribution.

For survival data (*Figure 5E*), the Mantel–Cox logrank test was employed for each pair-wise comparison between experimental groups. For the linear regression analysis presented in *Figure 4G*,

no constraints were applied as this was not deemed necessary, and no interpolation was performed because there was no missing data.

No data was excluded from any analysis, except in cases of sample attrition during processing, as occurred for one tumor sample during metabolomics processing in *Figure 5C*. For RT-qPCR experiments, if a single technical replicate for a given experimental condition did not show any amplification after 40 cycles, and if all other technical replicates for that experimental condition consistently showed amplification, then the non-amplification in the single replicate was attributed to random technical failure and that replicate was excluded. In instances where none of the technical replicates for a given experimental condition showed amplification after 40 cycles, but amplification was consistently observed in other experimental conditions assayed in parallel, the non-amplification for the mRNA of interest was attributed to the true absence of the mRNA from that sample, and the replicates were assigned a relative abundance value of zero.

## Acknowledgements

This work was supported in part by funding from the National Institutes of Health, including R01CA163649, R01CA270234, and U54CA274329 to PKS, F30CA265277 to NJM, R21CA251151 to AN, and R35GM126908 to DHP.

## Additional information

### Competing interests

David B Sykes: Co-founder and holds equity in Clear Creek Bio. The other authors declare that no competing interests exist.

### Funding

| Funder | Grant reference number | Author |
|---|---|---|
| National Cancer Institute | F30CA265277 | Nicholas J Mullen |
| National Cancer Institute | R01CA163649 | Pankaj K Singh |
| National Cancer Institute | U54CA274329 | Pankaj K Singh |
| National Cancer Institute | R21CA251151 | Amarnath Natarajan |
| National Institute of General Medical Sciences | R35GM126908 | David H Price |
| National Cancer Institute | R01CA270234 | Pankaj K Singh |

The funders had no role in study design, data collection and interpretation, or the decision to submit the work for publication.

### Author contributions

Nicholas J Mullen, Conceptualization, Data curation, Formal analysis, Funding acquisition, Validation, Investigation, Visualization, Methodology, Writing - original draft, Writing – review and editing; Surendra K Shukla, Dezhen Wang, Nina Chaika, Investigation, Methodology; Ravi Thakur, Data curation, Investigation, Methodology; Sai Sundeep Kollala, Validation, Investigation, Methodology; Juan F Santana, Formal analysis, Investigation, Methodology; William R Miklavcic, Validation, Investigation; Drew A LaBreck, Investigation; Jayapal Reddy Mallareddy, Resources; David H Price, Resources, Funding acquisition, Investigation, Methodology, Writing – review and editing; Amarnath Natarajan, Resources, Supervision; Kamiya Mehla, Resources, Methodology, Writing – review and editing; David B Sykes, Resources, Supervision, Writing – review and editing; Michael A Hollingsworth, Resources, Supervision, Funding acquisition, Project administration; Pankaj K Singh, Conceptualization, Resources, Data curation, Formal analysis, Supervision, Funding acquisition, Project administration, Writing – review and editing

## Author ORCIDs
Nicholas J Mullen http://orcid.org/0000-0002-5045-0814
Sai Sundeep Kollala http://orcid.org/0000-0002-9651-1723
Amarnath Natarajan http://orcid.org/0000-0001-5067-0203
Pankaj K Singh http://orcid.org/0000-0001-8903-0131

## Ethics
All procedures were approved by the Institutional Animal Care and Use Committee (IACUC) at the University of Nebraska Medical Center (protocol number: 20-112-03-FC).

Reviewer #1 (Public review): https://doi.org/10.7554/eLife.87292.3.sa1
Reviewer #2 (Public review): https://doi.org/10.7554/eLife.87292.3.sa2
Reviewer #3 (Public review): https://doi.org/10.7554/eLife.87292.3.sa3
Author response https://doi.org/10.7554/eLife.87292.3.sa4

---

# Additional files

## Supplementary files
• Supplementary file 1. Table listing the name, manufacturer, and catalog number of all small molecules used in this study. The concentration of each agent that was used for *Figure 2F* is also listed.

• Supplementary file 2. Table listing the gene symbol and NCBI gene ID for each gene that was named in this this article. For genes that were assayed by RT-qPCR, forward and reverse primer sequences are listed. 'N/A' indicates that primer sequence is not applicable because that gene was not assayed by RT-qPCR in this study.

• MDAR checklist

## Data availability
All data generated or analyzed in this study have been included in the manuscript and supporting files; source data files have been provided for all figures. Raw RNA sequencing data generated for this study have been deposited in the NCBI Sequence Read Archive (SRA) under the accession code PRJNA1099696. Sequencing experiment datasets from a previous publication *Tan et al., 2016* that were re-analyzed for this work were obtained via Gene Expression Omnibus (GEO) and the relevant accession numbers can be found in the Materials and methods section.

The following dataset was generated:

| Author(s) | Year | Dataset title | Dataset URL | Database and Identifier |
| --- | --- | --- | --- | --- |
| Singh PK | 2024 | S2-013 and CFPAC-1 cells were treated with brequinar for 16hr or 2 weeks at two different doses and bulk RNAseq was performed | https://www.ncbi.nlm.nih.gov/bioproject/?term=PRJNA1099696 | NCBI BioProject, PRJNA1099696 |

The following previously published dataset was used:

| Author(s) | Year | Dataset title | Dataset URL | Database and Identifier |
| --- | --- | --- | --- | --- |
| Zon L | 2016 | Nucleotide stress induction of HEXIM1 suppresses melanoma by modulating cancer cell-specific gene transcription | https://www.ncbi.nlm.nih.gov/geo/query/acc.cgi?acc=GSE68053 | NCBI Gene Expression Omnibus, GSE68053 |

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
