## [Editor Report · eLife assessment]

This **important** study reports a novel mechanism linking DHODH inhibition and subsequent pyrimidine nucleotide depletion with upregulation of cell surface MHC I in cancer cells. The in vitro mechanistic data are **compelling**, with rigorous methodology and validation across multiple cell lines. The authors also provide in vivo evidence for additive effects of DHODH inhibitors and immune checkpoint blockade. However, the in vivo assessments of the functional relevance of this mechanism remain **incomplete**, requiring additional analyses to fully substantiate the conclusions made.

---

## [Referee Report · Reviewer #1 (Public review)]

The manuscript by Mullen et al. investigated the gene expression changes in cancer cells treated with the DHODH inhibitor brequinar (BQ), to explore the therapeutic vulnerabilities induced by DHODH inhibition. The study found that BQ treatment causes upregulation of antigen presentation pathway (APP) genes and cell surface MHC class I expression, mechanistically which is mediated by the CDK9/PTEFb pathway triggered by pyrimidine nucleotide depletion. The combination of BQ and immune checkpoint therapy demonstrated a synergistic (or additive) anti-cancer effect against xenografted melanoma, suggesting the potential use of BQ and immune checkpoint blockade as a combination therapy in clinical therapeutics.

The interesting findings in the present study include demonstrating a novel cellular response in cancer cells induced by DHODH inhibition. However, whether the increased antigen presentation by DHODH inhibition actually contributed to the potentiation of the efficacy of immune-check blockade (ICB) is not directly examined is the limitation of the study. Moreover, the mechanism of the increased antigen presentation pathway by pyrimidine depletion mediated by CDK9/PTEFb was not validated by genetic KD or KO targeting by CDK9/PTEFb pathways. Finally, high concentrations of BQ have been reported to show off-target effects, sensitizing cancer cells to ferroptosis, and the authors should discuss whether the dose used in the in vivo study reached the ferroptotic sensitizing dose or not.

Comment on the revised version:

In their response letter, the authors appropriately addressed the reviewer's comments.

However, it is unfortunate that these comments are not reflected in the main text. Consequently, readers may encounter the same questions. Therefore, the reviewer recommends mentioning them in the discussion or limitations of the study, even if briefly, to address readers' concerns. Especially, addressing the comments such as the dosage of BQ being lower than the reported pro-ferroptotic dose (PMID 37407687), and the lack of examining potential impact of immune cell depletion on the efficacy of BQ treatment would be necessary for considering the proposed mechanism. The latter limitation is also raised by the other reviewer.

---

## [Referee Report · Reviewer #2 (Public review)]

In their manuscript entitled "DHODH inhibition enhances the efficacy of immune checkpoint blockade by increasing cancer cell antigen presentation", Mullen et al. describe an interesting mechanism of inducing antigen presentation. The manuscript includes a series of experiments that demonstrate that blockade of pyrimidine synthesis with DHODH inhibitors (i.e. brequinar (BQ)) stimulates the expression of genes involved in antigen presentation. The authors provide evidence that BQ mediated induction of MHC is independent of interferon signaling. A subsequent targeted chemical screen yielded evidence that CDK9 is the critical downstream mediator that induces RNA Pol II pause release on antigen presentation genes to increase expression. Finally, the authors demonstrate that BQ elicits strong anti-tumor activity in vivo in syngeneic models, and that combination of BQ with immune checkpoint blockade (ICB) results in significant lifespan extension in the B16-F10 melanoma model. Overall, the manuscript uncovers an interesting and unexpected mechanism that influences antigen presentation and provides an avenue for pharmacological manipulation of MHC genes, which is therapeutically relevant in many cancers. However, a few key experiments are needed to ensure that the proposed mechanism is indeed functional in vivo.

Major Points:

(1) According to the proposed model, BQ mediated induction of antigen presentation is a contributing factor to the efficacy of this therapeutic strategy. If this is true, then depletion of immune cells should reduce the therapeutic efficacy of BQ in vivo. The authors should perform the B16-F10 transplant experiments in either Rag null mice (if available) or with CD8/CD4 depletion. The expectation would be that T cell depletion (or MHC loss with genetic manipulation) should reduce the efficacy of BQ treatment. Absent this critical experiment, it is difficult to confidently conclude that induction of antigen presentation is a fundamental component of the in vivo response to DHODH inhibition.

(2) Does BQ treatment induce antigen presentation in non-malignant cells? APCs? If the induction of antigen presentation is not cancer specific and related to a pyrimidine depletion stress response, then there is a possibility that healthy tissues will also exhibit a similar phenotype, raising concerns about the specificity of a de novo immune response. The authors should examine antigen presentation genes in healthy tissues treated with BQ.

(3) In the title, the authors claim that DHODH enhances the efficacy of ICB. However, the experiment shown in Figure 5D does not demonstrate this. The Kaplan Meier curves reflect more of an additive response versus a synergistic combination. Furthermore, the concurrent treatment of BQ and ICB seems to inhibit the efficacy of ICB due to BQ toxicity in immune cells. When concurrently administered, the survival of the mice is the same as with brequinar alone, suggesting that the efficacy of ICB was diminished. However, if ICB is administered following an initial dose of BQ, there is an added survival benefit of a magnitude that is similar to ICB alone. This result seems to contradict the title. Furthermore, the authors should show the longitudinal growth curves of these tumors.

(4) Related to Point 3, the temporal separation of BQ and ICB raises the question of whether the induction of antigen presentation with BQ is persistent during the course of delayed ICB treatment. One explanation for the results is that BQ treatment reduces tumor burden, and then a subsequent course of ICB also reduces tumor burden but not that the two therapies are functioning in synergy. To address this, the authors should measure the duration of BQ mediated induction of antigen presentation after stopping treatment.

(5) In Figure 1, the authors show that DHODH inhibition induces expression of both MHC-I and MHC-II genes at the RNA level. However, they only validate MHC-I by flow cytometry. A simple experiment to evaluate the effect of BQ treatment on MHC-II surface expression would provide important additional mechanistic insight into the immunomodulatory effects of DHODH inhibition, especially given recent literature reinforcing the importance of MHC-II expression on epithelial cancers, including melanoma (Oliveira et al. Nature 2022).

Minor Points:

(1) The authors show ChIP-seq tracks from Tan et al. for HLA-B. However, given the pervasive effect of Ter treatment across many HLA genes, the authors should either show tracks at additional loci, or provide a heatmap of read density across more loci. This would substantiate the mechanistic claim that RNA Pol II occupancy and activity across antigen presentation genes is the major driver of response to DHODH inhibition as opposed to mRNA stabilization/increased translation.

(2) A compelling way to demonstrate a change in antigen presentation is through mass spectrometry based immunopeptidomics. Performing immunopeptidomic analysis of BQ treated cell lines would provide substantial mechanistic insight into the outcome of BQ treatment. While this approach may be outside the scope of the current work, the authors should speculate on how this treatment may specifically alter the antigenic landscape where future directions would include empirical immunopeptidomics measurements.

(3) While the signaling through CDK9 seems convincing, it still does not provide a mechanistic link between depleted pyrimidines and CDK9 activity. The authors should speculate on the mechanism that signals to CDK9.

(4) Related to minor point 2, the authors should consider a genetic approach to confirm the importance of CDK9. While the pharmacological approach, including multiple mechanistically distinct CDK9 inhibitors provides strong evidence, an additional experiment with genetic depletion of CDK9 (CRISPR KO, shRNA, etc) would provide compelling mechanistic confirmation.

(5) The authors should comment in the discussion on how this strategy may be particularly useful in patients harboring genetic or epigenetic loss of interferon signaling, a known mechanism of ICB resistance. Perhaps DHODH inhibition could rescue MHC expression in cells that are deficient in interferon sensing.

Overall, the paper is clearly written and presented. With the additional experiments described above, especially in vivo, this manuscript would provide a strong contribution to the field of antigen presentation in cancer. The distinct mechanisms by which DHODH inhibition induces antigen presentation will also set the stage for future exploration into alternative methods of antigen induction.

Comments on latest version:

The authors address the majority of the points raised in my previous review. However, no additional in vivo experiments were performed, which seems necessary for the major conclusions of the paper.

I disagree with the authors' assessment of Major Point 3 in my review. I have updated the text of Major Point 3 in my public review to further clarify my position.

My final assessment is that if the authors want to claim that DHODH inhibition potentiates immune checkpoint blockade, as is stated in the title, then further in vivo experimentation is needed.

---

## [Referee Report · Reviewer #3 (Public review)]

Mullen et al present an important study describing how DHODH inhibition enhances efficacy of immune checkpoint blockade by increasing cell surface expression of MHC I in cancer cells. DHODH inhibitors have been used in the clinic for many years to treat patients with rheumatoid arthritis and there has been a growing interest in repurposing these inhibitors as anti-cancer drugs. In this manuscript, the Singh group builds on their previous work defining combinatorial strategies with DHODH inhibitors to improve efficacy. The authors identify an increased expression of genes in the antigen presentation pathway and MHC I after BQ treatment which is mediated strictly by pyrimidine depletion and CDK9/P-TEFb. The authors rationalize that increased MHC I expression induced by DHODH inhibition might favor efficacy of dual immune checkpoint blockade. In fact, this combinatorial treatment prolonged survival in an immunocompetent B16F10 melanoma model.

Previous studies have shown that DHODH inhibitors can increase expression of innate immunity-related genes but the role of DHODH and pyrimidine nucleotides in antigen presentation has not been previously reported. A strength of the manuscript is the solid in vitro mechanistic data supported by analysis in multiple cell lines. The in vivo data show compelling additive effects of DHODH inhibitors and ICB. However, more controls and experiments would be required to define the nature of these effects and to confirm that the mechanistic in vitro data is conserved in vivo.

This is a relevant manuscript proposing a mechanistic link between pyrimidine depletion and MHC I expression and a novel therapeutic approach combining DHODH inhibitors with dual checkpoint blockade. These results might be relevant for the clinical development of DHODH inhibitors in the treatment of solid tumors, a setting where these have not shown optimal efficacy yet.

Comments on revised version:

The authors have addressed my questions regarding validation of gene expression in other cell lines. They have also provided an explanation about why in vivo evaluations could not be performed for the experiment in Figure 5E.

---

## [Author Response]

The following is the authors’ response to the previous reviews.

**eLife assessment**
This important study reports a novel mechanism linking DHODH inhibition-mediated pyrimidine nucleotide depletion to antigen presentation. Alternative means of inducing antigen presentation provide therapeutic opportunities to augment immune checkpoint blockade for cancer treatment. While the solid mechanistic data in vitro are compelling, in vivo assessments of the functional relevance of this mechanism are still incomplete.
**Public Reviews:**

We thank all Reviewers for their insightful comments and excellent suggestions.

**Reviewer #1 (Public Review):**
The manuscript by Mullen et al. investigated the gene expression changes in cancer cells treated with the DHODH inhibitor brequinar (BQ), to explore the therapeutic vulnerabilities induced by DHODH inhibition. The study found that BQ treatment causes upregulation of antigen presentation pathway (APP) genes and cell surface MHC class I expression, mechanistically which is mediated by the CDK9/PTEFb pathway triggered by pyrimidine nucleotide depletion.

No comment from authors

The combination of BQ and immune checkpoint therapy demonstrated a synergistic (or additive) anti-cancer effect against xenografted melanoma, suggesting the potential use of BQ and immune checkpoint blockade as a combination therapy in clinical therapeutics.

No comment from authors

The interesting findings in the present study include demonstrating a novel cellular response in cancer cells induced by DHODH inhibition. However, whether the increased antigen presentation by DHODH inhibition actually contributed to the potentiation of the efficacy of immune-check blockade (ICB) is not directly examined is the limitation of the study.

No comment from authors for preceding text, comment addresses the following text

Moreover, the mechanism of the increased antigen presentation pathway by pyrimidine depletion mediated by CDK9/PTEFb was not validated by genetic KD or KO targeting by CDK9/PTEFb pathways.

We appreciate this comment, and we would like to explain why we did not pursue these approaches. According to DepMap, CRISPR/Cas9-mediated knockout of CDK9 in cancer cell lines is almost universally deleterious, scoring as “essential” in 99.8% (1093/1095) of all cell lines tested (see Author response image 1 below). This makes sense, as P-TEFb is required for productive RNA polymerase II elongation of most mammalian genes. As such, it was not feasible to generate cell lines with stable genetic knockout of CDK9 to test our hypothesis.

While knockdown of CDK9 by RNA interference could support our results, DepMap data seems to indicate that RNAi-mediated knockdown of CDK9 is generally ineffective in silencing its activity, as this perturbation scored as “essential” in only 6.2% (44/710) of tested cell lines. This suggests that incomplete depletion of CDK9 will likely not be sufficient to block APP induction downstream of nucleotide depletion. Furthermore, RNAi-mediated depletion of CDK9 may trigger transcriptional changes in the cell by virtue of its many documented protein-protein interactions, and it would be difficult to establish a consistent “time zero” at which point CDK9 protein depletion is substantial but secondary effects of this have not yet occurred to a significant degree. These factors constitute major limitations of experiments using RNAi-mediated knockdown of CDK9.

**Author response image 1. sa4fig1:** Essentiality score from CRISPR and RNAi perturbation of CDK9 in cancer cell lines https://depmap.org/portal/gene/CDK9?tab=overview&dependency=RNAi_merged.

At any rate, we provide evidence that three different inhibitors of CDK9 (flavopiridol, dinaciclib, and AT7519) all inhibit our effect of interest (Fig 4B). The same results were observed using a previously validated CDK9-directed proteolysis targeting chimera (PROTAC2), and this was reversed by addition of excess pomalidomide (Fig 4C), which correlated with the presence/absence of CDK9 on western blot under the exact same conditions (Fig 4D).

It is formally possible that all CDK9 inhibitors we tested are blocking BQ-mediated APP induction by some shared off-target mechanism (or perhaps by two or more different off-target mechanisms) AND this CDK9-independent target also happens to be degraded by PROTAC2. However, this would be an extraordinarily non-parsimonious explanation for our results, and so we contend that we have provided compelling evidence for the requirement of CDK9 for BQ-mediated APP induction.

Finally, high concentrations of BQ have been reported to show off-target effects, sensitizing cancer cells to ferroptosis, and the authors should discuss whether the dose used in the in vivo study reached the ferroptotic sensitizing dose or not.

We are intrigued by the results shown to us by Reviewer #1 in the linked preprint (Mishima et al 2022, https://doi.org/10.21203/rs.3.rs-2190326/v1). We have also observed in our unpublished data that very high concentrations of BQ (>150µM) cause loss of cell viability that is not rescued by uridine supplementation and that occurs even in DHODH knockout cells. This effect of high-dose BQ must be DHODH-independent. We also agree that Mishima et al provide compelling evidence that the ferroptosis-sensitizing effect of high-dose BQ treatment is due (at least in large part) to inhibition of FSP1.

Although we showed that DHODH is strongly inhibited in tumor cells in vivo (Fig 5C), we did not directly measure the concentration of BQ in the tumor or plasma. Sykes et al (PMID: 27641501) found that the maximum plasma concentration (Cmax) for [BQ]free following a single IP administration in C57Bl6/J mice (15mg/kg) is approximately 3µM, while the Cmax for [BQ]total was around 215µM. Because polar drug molecules bound to serum proteins (predominantly albumin) are not available to bind other targets, [BQ]free is the relevant parameter.

Given a Cmax for [BQ]free of 3µM and half-life of 12.0 hours, we estimate that the steady-state [BQ]free with daily IP injections at this dose is around 4µM. Since we used an administration schedule of 10mg/kg every 24 hours, we estimate that the steady-state plasma [BQ]free in our system was 2.67µM (assuming initial Cmax of 2µM and half-life of 12.0 hours).

To derive an upper-bound estimate for the Cmax of [BQ]free over the 12-day treatment period (Fig 5A-D), we will use the observed data for 15mg/kg dose, and we will assume that (1) there is no clearance of BQ whatsoever and (2) that [BQ]free increases linearly with increasing [BQ]total. This yields a maximum free BQ concentration of 12 x 3 = 36µM.

Therefore, we consider it very unlikely that plasma concentrations of free BQ in our experiment exceeded the lower limit of the ferroptosis-sensitizing dose range reported by Mishima et al. However, without direct pharmacokinetic analysis, we cannot say for sure what the maximal [BQ]free was under our experimental conditions.

**Reviewer #2 (Public Review):**
In their manuscript entitled "DHODH inhibition enhances the efficacy of immune checkpoint blockade by increasing cancer cell antigen presentation", Mullen et al. describe an interesting mechanism of inducing antigen presentation. The manuscript includes a series of experiments that demonstrate that blockade of pyrimidine synthesis with DHODH inhibitors (i.e. brequinar (BQ)) stimulates the expression of genes involved in antigen presentation. The authors provide evidence that BQ mediated induction of MHC is independent of interferon signaling. A subsequent targeted chemical screen yielded evidence that CDK9 is the critical downstream mediator that induces RNA Pol II pause release on antigen presentation genes to increase expression. Finally, the authors demonstrate that BQ elicits strong anti-tumor activity in vivo in syngeneic models, and that combination of BQ with immune checkpoint blockade (ICB) results in significant lifespan extension in the B16-F10 melanoma model. Overall, the manuscript uncovers an interesting and unexpected mechanism that influences antigen presentation and provides an avenue for pharmacological manipulation of MHC genes, which is therapeutically relevant in many cancers. However, a few key experiments are needed to ensure that the proposed mechanism is indeed functional in vivo.The combination of DHODH inhibition with ICB reflects more of an additive response instead of a synergistic combination. Moreover, the temporal separation of BQ and ICB raises the question of whether the induction of antigen presentation with BQ is persistent during the course of delayed ICB treatment. To confidently conclude that induction of antigen presentation is a fundamental component of the in vivo response to DHODH inhibition, the authors should examine whether depletion of immune cells can reduce the therapeutic efficacy of BQ in vivo.

We concur with this assessment.

Moreover, they should examine whether BQ treatment induces antigen presentation in non-malignant cells and APCs to determine the cancer specificity.

Although we showed that this occurs in HEK-293T cells, we appreciate that this cell line is not representative of human cells of any organ system in vivo. So, we agree it is important to determine if DHODH inhibition induces antigen presentation in human tissues and professional antigen presenting cells, and this is an excellent focus for future studies.

However, it should also be noted that increased antigen presentation in non-malignant host tissues would not be expected to generate an autoimmune response, because host tissues likely lack strong neoantigens, and whatever immunogenic peptides they may have would likely be presented via MHC-I at baseline (i.e. even in the absence of DHODH inhibitor treatment), since all nucleated cells express MHC-I.

This argument is strongly supported by clinical experience/data, as DHODH inhibitors (leflunomide and teriflunomide) are commonly used to treat rheumatoid arthritis and multiple sclerosis. While the pathophysiology of these autoimmune syndromes is complex, it is thought that both diseases are driven by aberrant T-cell attack on host tissues, mediated by incorrect recognition of host antigens presented via MHC-I (as well as MHC-II) as “foreign.”

If increased antigen presentation in host tissues (downstream of DHODH inhibition) could lead to a de novo autoimmune response, then administration of DHODH inhibitors would be expected to exacerbate T-cell driven autoimmune disease rather than ameliorate it. Randomized controlled trials have consistently found that treatment with DHODH inhibitors leads to improvement of rheumatoid arthritis and multiple sclerosis symptoms, which is the opposite of what one would expect if DHODH inhibitors are causing de novo autoimmune reactions in human patients.

Finally, although the authors show that DHODH inhibition induces expression of both MHC-I and MHC-II genes at the RNA level, only MHC-I is validated by flow cytometry given the importance of MHC-II expression on epithelial cancers, including melanoma, MHC-II should be validated as well.

We fully agree with this statement. We attempted to quantify cell surface MHC-II expression by FACS using the same method as for MHC-I (Figs 1G-H, 2D, and 3F). We did not detect cell surface MHC-II in any of our cancer cell lines, despite the use of high-dose interferon gamma and other stimulants (which robustly increase MHC-II mRNA in our system) in an attempt to induce expression. However, because we did not use cells known to express MHC-II as a positive control (e.g. B-cell leukemia cell lines or primary splenocytes), we do not know if our results are due to some technical failure (perhaps related to our protocol/reagents) or if they reflect a true absence of cell surface MHC-II in our cell lines.

If the latter is true, that implies that either (1) MHC-II mRNA is not translated or (2) that it is translated, but our cancer cell lines lack one or more elements of the machinery required for MHC-II antigen presentation.

In any case, it is important to determine if DHODH inhibition increases MHC-II at the cell surface of cancer cells using appropriate positive and negative controls, as this could have important implications for cancer immunotherapy.

[As a minor point, melanoma is not an epithelial cancer, as it is derived from neural crest lineage cells (melanocytes)]

Overall, the paper is clearly written and presented. With the additional experiments described above, especially in vivo, this manuscript would provide a strong contribution to the field of antigen presentation in cancer. The distinct mechanisms by which DHODH inhibition induces antigen presentation will also set the stage for future exploration into alternative methods of antigen induction.
**Reviewer #3 (Public Review):**
Mullen et al present an important study describing how DHODH inhibition enhances efficacy of immune checkpoint blockade by increasing cell surface expression of MHC I in cancer cells. DHODH inhibitors have been used in the clinic for many years to treat patients with rheumatoid arthritis and there has been a growing interest in repurposing these inhibitors as anti-cancer drugs. In this manuscript, the Singh group build on their previous work defining combinatorial strategies with DHODH inhibitors to improve efficacy. The authors identify an increase in expression of genes involved in the antigen presentation pathway and MHC I after BQ treatment and they narrow the mechanism to be strictly pyrimidine and CDK9/P-TEFb dependent. The authors rationalize that increased MHC I expression induced by DHODH inhibition might favor efficacy of dual immune checkpoint blockade. This combinatorial treatment prolonged survival in an immunocompetent B16F10 melanoma model.

[No comment from authors]

Previous studies have shown that DHODH inhibitors can increase expression of innate immunity-related genes but the role of DHODH and pyrimidine nucleotides in antigen presentation has not been previously reported. A strength of the manuscript is the use of multiple controls across a panel of cell lines to exclude off-target effects and to confirm that effects are exclusively dependent on pyrimidine depletion. Overall, the authors do a thorough characterization of the mechanism that mediates MHC I upregulation using multiple strategies. Furthermore, the in vivo studies provide solid evidence for combining DHODH inhibitors with immune checkpoint blockade.

No comment from authors

However, despite the use of multiple cell lines, most experiments are only performed in one cell line, and it is hard to understand why particular gene sets, cell lines or time points are selected for each experiment. It would be beneficial to standardize experimental conditions and confirm the most relevant findings in multiple cell lines.

We appreciate this comment, and we understand how the use of various cell lines may seem puzzling. We would like to explain how our cell line panel evolved over the course of the study.Our first indication that BQ caused APP upregulation came from transcriptomics experiments (Figs 1A-D, S1A) performed as part of a previous study investigating BQ resistance (Mullen et al, 2023 Cancer Letters). In that study, we used CFPAC-1 as a model for BQ sensitivity and S2-013 as a model for BQ resistance. We did RNA sequencing +/- BQ in these cell lines to look for gene expression patterns that might underlie resistance/sensitivity to BQ. When analyzing this data, we serendipitously discovered the APP/MHC phenomenon, which gave rise to the present study.

Our next step was to extend these findings to cancer cell lines of other histologies, and we prioritized cell lines derived from common cancer types for which immunotherapy (specifically ICB) are clinically approved. This is why A549 (lung adenocarcinoma), HCT116 (colorectal adenocarcinoma), A375 (cutaneous melanoma), and MDA-MB-231 (triple-negative breast cancer) cell lines were introduced.

Because PDAC is considered to have an especially “immune-cold” tumor microenvironment, we reasoned that even dramatically increasing cancer cell antigen presentation may be insufficient to elicit an effective anti-tumor immune response in vivo. So we shifted our focus towards melanoma, because a subset of melanoma patients is very responsive to ICB and loss of antigen presentation (by direct silencing or homozygous loss-of-function mutations in MHC-I components such as B2M, or by functional loss of IFN-JAK1/2-STAT signaling) has been shown to mediate ICB resistance in human melanoma patients. This is why we extended our findings to B16F10 murine melanoma cells, intending to use them for in vivo studies with syngeneic immunocompetent recipient mice.

The PDAC cell line MiaPaCa2 was introduced because a collaborator at our institution (Amar Natarajan) happened to have IKK2 knockout MiaPaCa2 cells, which allowed us to genetically validate our inhibitor results showing that IKK1 and IKK2 (crucial effectors for NF-kB signaling) are dispensable for our effect of interest.

Ultimately, realizing that our results spanned various human and murine cell lines, we chose to use HEK-293T cells to validate the general applicability of our findings to proliferating cells in 2D culture, since HEK-293T cells (compared to our cancer cell lines) have relatively few genetic idiosyncrasies and express MHC-I at baseline.

The differential in vivo survival depending on dosing schedule is interesting. However, this section could be strengthened with a more thorough evaluation of the tumors at endpoint.Overall, this is an interesting manuscript proposing a mechanistic link between pyrimidine depletion and MHC I expression and a novel therapeutic strategy combining DHODH inhibitors with dual checkpoint blockade. These results might be relevant for the clinical development of DHODH inhibitors in the treatment of solid tumors, a setting where these inhibitors have not shown optimal efficacy yet.
**Recommendations for the authors:**

**Reviewer #1 (Recommendations For The Authors):**
(1) The main issue is that it did not directly examine whether the increased antigen presentation by DHODH inhibition contributed to the potentiation of the efficacy of immune-check blockade (ICB). The additional effect of BQ in the xenograft tumor study was not examined to determine if it was due to increased antigen presentation toward the cancer cells or due to merely cell cycle arrest effect by pyrimidine depletion in the tumor cells. The different administration timing of ICB with BQ treatment (Fig 5E) would not be sufficient to answer this issue.

We agree with this assessment and, and we believe the experiment proposed by Reviewer #2 below (comparing the efficacy of BQ in Rag-null versus immunocompetent recipients) would address this question directly. We also think that using a more immunogenic cell line for this experiment (such as B16F10 transduced with ovalbumin or some other strong neoantigen) would be useful given the poor immunogenicity and lack of any defined strong neoantigen in B16F10 cells. An orthogonal approach would be to engraft cancer cells with or without B2M knockout into immunocompetent recipient mice (+/- BQ treatment) to further implicate MHC-I and antigen presentation. These questions will be addressed in future studies.

(2) Additionally, in the in vivo study, the increase in surface MHC1 in the protein level in by BQ treatment was not examined in the tumor samples, and it was not confirmed whether increased antigen presentation by BQ treatment actually promoted an anti-cancer immune response in immune cells. To support the story presented in the study, these data would be necessary.

We attempted to show this by immunohistochemistry, but unfortunately the anti-H2-Db antibody that we obtained for this purpose did not have satisfactory performance to assess this in our tissue samples harvested at necropsy.

(3) The mechanism of the increased antigen presentation pathway by pyrimidine depletion mediated by CDK9/PTEFb was not validated by genetic KD or KO targeting by CDK9/PTEFb pathways. In general, results only by the inhibitor assay have a limitation of off-target effects.

Please see our above reply to Reviewer #1 comment making this same point, where we spell out our rationale for not pursuing these experiments.

(4) High concentrations of BQ (> 50 uM) have been reported to show off-target effects, sensitizing cancer cells to ferroptosis, an iron-mediated lipid peroxidation-dependent cell death, independent of DHODH inhibition (https://www.researchsquare.com/article/rs-2190326/v1). It would be needed to discuss whether the dose used in the in vivo study reached the ferroptotic sensitizing dose or not.

Please see our above reply to Reviewer #1 comment making this same point, where we explain why we are very confident that the BQ dose administered in our animal experiments was far below the minimum reported BQ dose required to sensitize cancer cells to ferroptosis in vitro.

**Reviewer #2 (Recommendations For The Authors):**
Major Points(1) According to the proposed model, BQ mediated induction of antigen presentation is a contributing factor to the efficacy of this therapeutic strategy. If this is true, then depletion of immune cells should reduce the therapeutic efficacy of BQ in vivo. The authors should perform the B16-F10 transplant experiments in either Rag null mice (if available) or with CD8/CD4 depletion. The expectation would be that T cell depletion (or MHC loss with genetic manipulation) should reduce the efficacy of BQ treatment. Absent this critical experiment, it is difficult to confidently conclude that induction of antigen presentation is a fundamental component of the in vivo response to DHODH inhibition.

We agree with this assessment and the proposed experiment comparing the response in Rag-null versus immunocompetent recipients. We also think that using a more immunogenic cell line for this experiment (such as B16F10 transduced with ovalbumin or some other strong neoantigen) would be useful given the poor immunogenicity and lack of any defined strong neoantigen in B16F10 cells. An orthogonal approach would be to engraft cancer cells with or without B2M knockout into immunocompetent recipient mice (+/- BQ treatment) to further implicate MHC-I and antigen presentation. These questions will be addressed in future studies.

(2) Does BQ treatment induce antigen presentation in non-malignant cells? APCs? If the induction of antigen presentation is not cancer specific and related to a pyrimidine depletion stress response, then there is a possibility that healthy tissues will also exhibit a similar phenotype, raising concerns about the specificity of a de novo immune response. The authors should examine antigen presentation genes in healthy tissues treated with BQ.

We agree it is important to examine if our findings regarding nucleotide depletion and antigen presentation are true of APCs and other non-transformed cells, but we are not so concerned about the possibility of raising an immune response against non-malignant host tissues, as explained above. We have reproduced the relevant section below:

“However, it should also be noted that increased antigen presentation in non-malignant host tissues would not be expected to generate an autoimmune response, because host tissues likely lack strong neoantigens, and whatever immunogenic peptides they may have would likely be presented via MHC-I at baseline, since all nucleated cells express MHC-I.

This argument is strongly supported by clinical experience/data, as DHODH inhibitors (leflunomide and teriflunomide) are commonly used to treat rheumatoid arthritis and multiple sclerosis. While the pathophysiology of these autoimmune syndromes is complex, it is thought that both diseases are driven by aberrant T-cell attack on host tissues, mediated by incorrect recognition of host antigens presented via MHC-I (as well as MHC-II) as “foreign.”

If increased antigen presentation in host tissues (downstream of DHODH inhibition) could lead to a de novo autoimmune response, then administration of DHODH inhibitors would be expected to exacerbate T-cell driven autoimmune disease rather than ameliorate it. Randomized controlled trials have consistently found that treatment with DHODH inhibitors leads to improvement of rheumatoid arthritis and multiple sclerosis symptoms, which is the opposite of what one would expect if DHODH inhibitors are causing de novo autoimmune reactions in human patients.”

(3) In the title, the authors claim that DHODH enhances the efficacy of ICB. However, the experiment shown in Figure 5D does not demonstrate this. The Kaplan Meier curves reflect more of an additive response versus a synergistic combination. Furthermore, the concurrent treatment of BQ and ICB seems to inhibit the efficacy of ICB due to BQ toxicity in immune cells. This result seems to contradict the title.

We do not agree with this assessment. Given that the effect of dual ICB alone was very marginal, while the effect of BQ monotherapy was quite marked, we cannot conclude from Fig 5 that BQ treatment inhibited ICB efficacy due to immune suppression.

(4) Related to Point 3, the temporal separation of BQ and ICB raises the question of whether the induction of antigen presentation with BQ is persistent during the course of delayed ICB treatment. One explanation for the results is that BQ treatment reduces tumor burden, and then a subsequent course of ICB also reduces tumor burden but not that the two therapies are functioning in synergy. To address this, the authors should measure the duration of BQ mediated induction of antigen presentation after stopping treatment.

We agree that the alternative explanation proposed by Reviewer #2 is possible and we appreciate the suggestion to test the stability of APP induction after stopping BQ treatment.

(5) In Figure 1, the authors show that DHODH inhibition induces expression of both MHC-I and MHC-II genes at the RNA level. However, they only validate MHC-I by flow cytometry. A simple experiment to evaluate the effect of BQ treatment on MHC-II surface expression would provide important additional mechanistic insight into the immunomodulatory effects of DHODH inhibition, especially given recent literature reinforcing the importance of MHC-II expression on epithelial cancers, including melanoma (Oliveira et al. Nature 2022).

We fully agree with this statement. We attempted to quantify cell surface MHC-II expression by FACS using the same method as for MHC-I (Figs 1G-H, 2D, and 3F). We did not detect cell surface MHC-II in any of our cancer cell lines, despite the use of high-dose interferon gamma and other stimulants (which robustly increase MHC-II mRNA in our system) in an attempt to induce expression. However, because we did not use cells known to express MHC-II as a positive control (e.g. B-cell leukemia cell lines or primary splenocytes), we do not know if our results are due to some technical failure (perhaps related to our protocol/reagents) or if they reflect a true absence of cell surface MHC-II in our cell lines.

If the latter is true, that implies that either (1) MHC-II mRNA is not translated or (2) that it is translated, but our cancer cell lines lack one or more elements of the machinery required for MHC-II antigen presentation.

In any case, it is important to determine if DHODH inhibition increases MHC-II at the cell surface of cancer cells using appropriate positive and negative controls, as this could have important implications for cancer immunotherapy.

[As a minor point, melanoma is not an epithelial cancer, as it is derived from neural crest lineage cells (melanocytes)]

Minor Points(1) The authors show ChIP-seq tracks from Tan et al. for HLA-B. However, given the pervasive effect of Ter treatment across many HLA genes, the authors should either show tracks at additional loci, or provide a heatmap of read density across more loci. This would substantiate the mechanistic claim that RNA Pol II occupancy and activity across antigen presentation genes is the major driver of response to DHODH inhibition as opposed to mRNA stabilization/increased translation.

We appreciate this suggestion. We have changed Fig 4 by replacing the HLA-B track (old Fig 4E) with a representation of fold change (Ter/DMSO) in Pol II occupancy versus fold change (Ter/DMSO) in mRNA abundance for 23 relevant genes (new Fig 4G); both of these datasets were obtained from the Tan et al manuscript. This new figure panel (Fig 4G) also shows linear regression analysis demonstrating that Pol II occupancy and mRNA expression are significantly correlated for APP genes. While we recognize that this data in itself is not formal proof of our hypothesis, it does strongly support the notion that increased transcription is responsible for the increased mRNA abundance of APP genes that we have observed.

(2) A compelling way to demonstrate a change in antigen presentation is through mass spectrometry based immunopeptidomics. Performing immunopeptidomic analysis of BQ treated cell lines would provide substantial mechanistic insight into the outcome of BQ treatment. While this approach may be outside the scope of the current work, the authors should speculate on how this treatment may specifically alter the antigenic landscape where future directions would include empirical immunopeptidomics measurements.

We fully agree with this comment. While the abundance of cancer cell surface MHC-I is an important factor for anticancer immunity, another crucial factor is the identity of peptides that are presented. Treatments that cause presentation of more immunogenic peptides can enhance T-cell recognition even in the absence of a relative change in cell surface MHC-I abundance.

While we did not perform the immunopeptidomics experiments described, we can offer some speculation regarding this comment. As shown in Fig 1D-E, transcriptomics experiments suggest that immunoproteasome subunits (PSMB8, PSMB9, PSMB10) are upregulated upon DHODH inhibition. If this change in mRNA levels translates into greater immunoproteasome activity (which was not tested in our study), this would be expected to alter the repertoire of peptides available for presentation and could thereby change the immunopeptidome.

However, this hypothesis requires direct testing, and we hope future studies will delineate the effects of DHODH inhibition and other cancer therapies on the immunopeptidome, as this area of research will have important clinical implications.

(3) While the signaling through CDK9 seems convincing, it still does not provide a mechanistic link between depleted pyrimidines and CDK9 activity. The authors should speculate on the mechanism that signals to CDK9.We agree with the assessment. A mechanistic link between depleted pyrimidines and CDK9 activity will be a subject of future studies.(4) Related to minor point 2, the authors should consider a genetic approach to confirm the importance of CDK9. While the pharmacological approach, including multiple mechanistically distinct CDK9 inhibitors provides strong evidence, an additional experiment with genetic depletion of CDK9 (CRISPR KO, shRNA, etc) would provide compelling mechanistic confirmation.

Reviewer #1 raised this very same point, and we agree. Please see our reply to Reviewer #1, which details why we did not pursue this approach and argues that the evidence we present is compelling even in absence of genetic manipulation.

Additionally, please see the new Fig 4E and 4F, which is a repeat of Fig 4B using HCT116 cells. Figure 4E shows that, in this cell line, CDK9 inhibitors (flavopiridol, dinaciclib, and AT7519) block BQ-mediated APP induction, while PROTAC2 does not. Figure 4F shows that (for reasons we cannot fully explain) PROTAC2 does not lead to CDK9 degradation in HCT116 cells. This data strongly implicates CDK9, because it excludes a CDK9-degradation-independent effect of PROTAC2.

(5) Figure 2B needs a legend.

Thank you for pointing this out. We have added a legend to Fig 2B.

(6) The authors should comment in the discussion on how this strategy may be particularly useful in patients harboring genetic or epigenetic loss of interferon signaling, a known mechanism of ICB resistance. Perhaps DHODH inhibition could rescue MHC expression in cells that are deficient in interferon sensing.

Thank you for this suggestion! We have amended the Discussion section to mention this important point. Please see paragraph 2 of the revised Discussion section where we have added the following text:

“Because BQ-mediated APP induction does not require interferon signaling, this strategy may have particular relevance for clinical scenarios in which tumor antigen presentation is dampened by the loss or silencing of cancer cell interferon signaling, which has been demonstrated to confer both intrinsic and acquired ICB resistance in human melanoma patients.”

**Reviewer #3 (Recommendations For The Authors):**
The authors present convincing evidence of the mechanism by which pyrimidine nucleotides regulate MHC I levels and about the potential of combining DHODH inhibitors with dual immune checkpoint blockade (ICB). This is an interesting paper given the clinical relevance of DHODH inhibitors. The studies raise some questions, and some points might need clarifying as below:In Figure 2C, why do the authors focus on these two genes in the uridine rescue? These are important genes mediating antigen presentation, but it might be more interesting to see how H2-Db and H2-Kb expression correlate with the protein data shown in Fig 2D. Fig. 2C-2D is a relevant control, so it would be important to validate in a different cancer cell line (e.g. one of the PDAC cell lines used for the RNAseq).

We appreciate this comment. Although Fig 3C shows that BQ-induced expression of H2-Db, H2-Kb, and B2m is reversed by uridine (in B16F10 cells), we recognize that this was not the best placement for this data, as it can easily be overlooked here since uridine reversal is not the main point of Fig 3C. We have left Fig 3C as is, because we think that the uridine reversal demonstrated in that panel serves as a good internal positive control for reversal of BQ-mediated APP induction in that experiment.

We have repeated the experiments shown in the original Fig 2C and substituted the original Fig 2C with a new Fig 2C and Fig S2B, which show both Tap1 and Nlrc5 as well as H2-Db, H2-Kb, and B2m after treatment with either BQ (new Fig 2C) or teriflunomide (new Fig S2B). The original Fig S2B is now Fig S2C, and it shows that uridine has no effect on the expression of any of the genes assayed in the new Fig 2C or S2B.

The reversibility of cell surface MHC-I induction was also validated in HCT116 cells (Fig 3F). We included the uridine reversal in Fig 3F to avoid duplicating the control and BQ FACS data in multiple panels.

We have also added the qPCR data for HCT116 cells showing this same phenotype (at the mRNA level), which is the new Fig S2D.

We decided to prioritize HCT116 cells for our mechanistic studies (Figures S2D, S4A, and 4E-F) because previous reports indicate that it is diploid and therefore less genetically deranged compared to our other cancer cell lines.

Figure 2F shows an elegant experiment to discard off-target effects related to cell death and to confirm that the increased MHC I expression is uniquely dependent on pyrimidines. DHODH has recently been involved in ferroptosis, a highly immunogenic type of cell death. What are the authors´ thoughts on BQ-induced ferroptosis as a possible contributor to the effects of ICB? Does BQ + ferroptosis inhibitor (ferrostatin) affect cell surface MHC I and/or expression of antigen processing genes?

The potential role of DHODH in ferroptosis protection (Mao et al 2021) has important implications, so we are glad that multiple reviewers raised questions concerning ferroptosis. We did not directly test the effect of ferroptosis inducing agents (with or without BQ) on MHC-I/APP expression, but that is certainly a worthwhile line of investigation.

The DHODH/ferroptosis issue is complicated by a study pointed out by Reviewer #1 that challenges the role of DHODH inhibition in BQ-mediated ferroptosis sensitization (Mishima et al, 2022). This study argues that high-dose BQ treatment causes FSP1 inhibition, and this underlies the effect of BQ on the cellular response to ferroptosis-inducing agents.

Regardless of whether BQ-induced ferroptosis-sensitization is dependent on DHODH, FSP1, or some other factor, the Mao and Mishima studies agree that a relatively high dose of BQ is required to observe these effects (100-200µM for most cell lines and >50µM even in the most ferroptosis-sensitive cell lines). As we explained above, we consider it very unlikely that the in vivo BQ exposure in our experiments (Fig 5) was high enough to cause significant ferroptosis, especially in the absence of any dedicated ferroptosis-inducing agent (which is typically required to cause ferroptosis even in the presence of high-dose BQ).

The authors nail down the mechanism to CDK9 (Fig 4). However, all these experiments are performed in 293T cells. I would like to see a repeat of Fig. 4B in a cancer cell line (either PDAC or B16). Also, does BQ have any effect on CDK9 expression/protein levels?

We have added two figure panels that address this comment (new Fig 4E and 4F). Figure 4E (which is a repeat of Fig 4B with HCT116 cells) shows that CDK9 inhibitors (flavopiridol, AT7519, and dinaciclib) reverse BQ-mediated APP induction in HCT116 cells (this agrees with Fig S4A showing that flavopiridol reverses MHC induction by various nucleotide synthesis inhibitors in this cell line), but PROTAC2 does not. Figure 4F shows that PROTAC2 (for reasons we cannot explain) does not cause CDK9 degradation in HCT116 cells. This adds further support to our thesis that CDK9 is a critical mediator of BQ-mediated APP induction (because how else can this pattern of results be explained?). The text of the Results section has been amended to reflect this.

We chose to use HCT116 cells for this repeat experiment (1) to align with Fig S4A and (2) because, as previously mentioned, we consider HCT116 to be a good cell line for mechanistic studies because of its relative lack of idiosyncratic genetic features (compared to CFPAC-1, for example, which was derived from a patient with cystic fibrosis).

What are the differences in tumor size for the experiment shown in Figure 5E? What about tumor cell death in the ICB vs. BQ+ICB groups?

Because this was a survival assay, direct comparisons of tumor volumes between groups was not possible at later time points, since mice that die or have to be euthanized are removed from their experimental group, which lowers the average group tumor burden at subsequent time points. Although tumor volume was the most common euthanasia criteria reached, a subset of mice were either found dead or had to be euthanized for other reasons attributed to their tumor burden (moribund state, inability to ambulate or stand, persistent bleeding from tumor ulceration, severe loss of body mass, etc.). This confounds any comparison of endpoint measurements (such as immunohistochemical quantification of tumor cell death markers, T-cell markers, etc.).

The different response in the concurrent vs delayed treatment is very interesting. The authors suggest two possible mechanisms to explain this: "(1) Concurrent BQ dampens the initial anticancer immune response generated by dual ICB, or (b) cancer cell MHC-I and related genes are not maximally upregulated at the time of ICB administration with concurrent treatment". However, and despite the caveat of comparing the in vitro to the in vivo setting, Fig 2D shows upregulation of MHC I already at 24h of treatment in B16 cells. Have the authors checked T cell infiltration in the concurrent and delayed treatment setting?

For the same reasons described in response to the preceding comment, tumors harvested upon mouse death/euthanasia from our survival experiment were not suitable for cross-cohort comparison of tumor endpoint measurements. An additional experiment in which mice are necropsied at a prespecified time point (before any mice have died or reached euthanasia criteria, as in the experiment for Fig 5A-D) would be required to answer this question.

Page 5, line 181 -do the authors mean "nucleotide salvage inhibitors" instead of "synthesis"?

We believe the reviewer is referring to the following sentence:

“The other drugs screened included nucleotide synthesis inhibitors (5-fluorouracil, methotrexate, gemcitabine, and hydroxyurea), DNA damage inducers (oxaliplatin, irinotecan, and cytarabine), a microtubule targeting drug (paclitaxel), a DNA methylation inhibitor (azacytidine), and other small molecule inhibitors (Fig 2F).”

In this context, we believe our use of “synthesis” instead of “salvage” is correct, because methotrexate and 5-FU inhibit thymidylate synthase (which mediates de novo dTTP synthesis), while gemcitabine and hydroxyurea inhibit ribonucleotide reductase (which mediates de novo synthesis of all dNTPs).